# MiniMax-Remover: Taming Bad Noise Helps Video Object Removal

**Bojia Zi**[1,6,*], **Weixuan Peng**[2,*], **Xianbiao Qi**[3,†], **Jianan Wang**[4],
**Shihao Zhao**[5], **Rong Xiao**[3], **Kam-Fai Wong**[1,6]

[1]The Chinese University of Hong Kong    [2]Shenzhen University    [3]IntelliFusion Inc.

[4]Astribot Inc.    [5]The University of Hong Kong

[6]MoE Key Laboratory of High Confidence Software Technologies, CUHK

Figure 1: Visual Results of MiniMax-Remover. The left side displays the original videos, while the right side shows the edited results. Our method achieves high-quality removal of the target objects: the girl, chameleon, bird, lane line, and red wine glass, as illustrated in the five corresponding video examples. *Best viewed with Acrobat Reader. Click the images to play the animations.*

## Abstract

Recent advances in video diffusion models have driven rapid progress in video editing techniques. However, video object removal, a critical subtask of video editing, remains challenging due to issues such as hallucinated objects and visual artifacts. Furthermore, existing methods often rely on computationally expensive sampling procedures and classifier-free guidance (CFG), resulting in slow inference. To address these limitations, we propose **MiniMax-Remover**, a novel two-stage

* Equal Contribution.

† Corresponding author. Email: qixianbiao@gmail.com

39th Conference on Neural Information Processing Systems (NeurIPS 2025).

video object removal approach. Motivated by the observation that text condition is not best suited for this task, we simplify the pretrained video generation model by removing textual input and cross-attention layers. In this way, we obtain a more lightweight and efficient model architecture in the first stage. In the second stage, we proposed a minimax optimization strategy to further distill the remover with the successful videos produced by stage-1 model. Specifically, the inner maximization identifies adversarial input noise ("bad noise") that leads to failure removals, while the outer minimization trains the model to generate high-quality removal results even under such challenging conditions. As a result, our method achieves a state-of-the-art video object removal results using as few as 6 sampling steps without CFG usage. Extensive experiments demonstrate the effectiveness and superiority of MiniMax-Remover compared to existing methods. Codes and Videos are available at: `https://minimax-remover.github.io`.

# 1 Introduction

In recent years, diffusion models [4, 3, 16, 8, 11, 37, 51, 22, 34, 33, 44, 57] have driven remarkable progress in image and video generation. UNet-based architectures, exemplified by models such as AnimateDiff [16] and VideoCrafter2 [8], have demonstrated impressive capabilities in generating high-quality videos. More recently, the field has advanced with the introduction of transformer-based architectures, as seen in groundbreaking works such as Sora [37], HunyuanVideo [22] and Wan2.1 [44], which employ DiT-based structures [38] to achieve unprecedented generation quality. Parallel to these advancements, diffusion-based editing techniques [6, 52, 47, 12, 30, 48, 2, 20, 58, 28, 49, 13, 10, 23] have also seen significant progress. Among these, *video inpainting* has emerged as a particularly crucial capability, serving as a foundational component for numerous fine-grained video editing applications.

Video object removal focuses on erasing specified objects from videos while preserving background consistency and temporal coherence. ProPainter [56] addressed this by completing optical flow within masked regions and then synthesizing the fill-in content using vision transformers, rather than relying on generative models. More recently, the field has shifted toward diffusion-based methods, with several notable advancements. FFF-VDI [24] uses optical flow and generative model to inpaint videos. DiffuEraser [25] employs DDIM inversion and vision priors to enhance video inpainting quality. Similarly, FloED [15] proposes a dual-branch architecture and a two-stage training strategy to achieve higher fidelity in the regions where objects are intended to be removed.

Despite rapid progress, existing video object removal methods still face multiple challenges. Some methods generate undesired content or artifacts, while others introduce occlusions and blurs within masked regions. Furthermore, most approaches rely heavily on auxiliary priors (*e.g.*, optical flow, text prompts, or DDIM inversion). The complicated model design lead to unsuitability and limit user-friendliness. Moreover, the generative methods require a large number of sampling steps to achieve high visual fidelity and depend on classifier-free guidance (CFG), which requires two model evaluations per sampling step, further exacerbates inference latency, limiting their practicality in real-world applications.

To address these limitations, we propose a two-stage framework for video object removal that simultaneously enhances visual quality and reduces inference cost. In **Stage 1**, we train upon Wan2.1-1.3B [44] and introduce two key modifications: (i) replacing external text prompts with learnable *contrastive condition tokens*, and (ii) removing all cross-attention layers within the DiT blocks, instead injecting contrastive condition tokens directly into the self-attention stream to enable conditional control. This lightweight conditioning approach significantly reduces computational overhead while preserving the structural integrity of the DiT backbone, allowing the same weights to be reused in the next stage, even when conditional control is disabled. In **Stage 2**, we distill the remover on manually curate 10K high-quality object removal samples generated by the model in Stage 1 and introduce a *minimax* optimization strategy to further improve object removal performance. The inner maximization searches for adversarial noise ("bad" noise) that induces model failure, while the outer minimization trains the model to be robust against such challenging cases. This minmax optimization enables our remover to consistently produce artifact-free results and prevents generating undesired objects without the assistance of CFG. As a result, the final model achieves both superior visual quality and remarkable inference efficiency compared to existing methods.

Table 1: Comparison of our MiniMax-Remover and other Video Inpainting Methods. We compare these methods on a video with 33 frames and 360P resolution on A800 GPUs, using their default configuration. "-" means unknown, while "N/A" means not applicable. Senorita-R means Senorita-2M remover expert. DiffuEraser uses ProPainter to generate prior (1 step) and PCM [45] to refine the edited output (2 steps). The motion adapter (20 steps) and DDIM inversion in DiffuEraser are both optional.

| Method | Params | •Inference Details | | | | •Additional Condition / Prior | | | | Open-Source |
|---|---|---|---|---|---|---|---|---|---|---|
| | | Min Steps | CFG | Latency (s) | GPU Mem (GB) | DDIM Inversion | Optical Flow | Text Prompt | First Frame | |
| AVID [55] | 1.95B | - | ✓ | - | - | × | × | ✓ | × | × |
| FFF-VDI [24] | - | 50 | ✓ | - | - | ✓ | ✓ | × | × | × |
| VIVID [20] | 5.57B | 50 | ✓ | - | - | × | × | ✓ | ✓ | × |
| Senorita-R [58] | 1.69B | 50 | ✓ | - | - | × | × | ✓ | × | × |
| MTV-Inpaint [50] | 1.41B | 30 | ✓ | - | - | × | × | ✓ | × | × |
| Propainter [56] | 39.4M | N/A | N/A | 0.20 | 12.1 | × | ✓ | × | × | ✓ |
| VideoComposer [48] | 1.40B | 50 | ✓ | 2.83 | 49.1 | × | × | ✓ | × | ✓ |
| COCOCO [59] | 1.55B | 50 | ✓ | 3.56 | 36.5 | × | × | ✓ | × | ✓ |
| FloED [15] | 1.41B | 25 | ✓ | 1.32 | 47.6 | × | ✓ | ✓ | ✓ | ✓ |
| DiffuEraser [25] | 2.19B | 3 | ✓ | 0.35 | 10.4 | ✓ | × | ✓ | × | ✓ |
| VideoPainter [2] | 5.92B | 50 | ✓ | 8.14 | 44.7 | × | × | ✓ | ✓ | ✓ |
| VACE [21] | 1.78B | 25 | ✓ | 1.93 | 23.6 | × | × | ✓ | × | ✓ |
| **MiniMax-Remover** | 1.12B | 6 | × | 0.18 | 8.2 | × | × | × | × | ✓ |

**Our main contributions are summarized as follows:**

1. we propose a lightweight yet effective DiT-based architecture for video object removal. Motivated by the observation that text prompting is not best suited for the task of object removal, we replace the text conditions with learnable contrastive tokens to control the removal process. These tokens are integrated directly into the self-attention stream, allowing us to remove all cross-attention layers within the pretrained video generation model. *As a result, in Stage 1, our model features fewer parameters and no longer depends on ambiguous text instruction.*

2. In Stage 2, we distilled a remover on 10K manually selected video removals produced by the model in Stage 1, with minmax optimization strategy. Specifically, the inner maximization optimization finds a bad input noise that makes model fail, while the outer minimization optimization tries to teach model remove the bad noise added on the successful videos.

3. We conduct extensive experiments across multiple benchmarks, demonstrating that our method achieves superior performance in both inference speed and visual fidelity. As shown in Figure 4 and Table 1, *our model produces high-quality removal results using as few as 6 sampling steps and without relying on classifier-free guidance (CFG).*

## 2 Related Work

Video diffusion models have witnessed remarkable progress in recent years [48, 46, 16, 8, 37, 51, 33, 22, 17, 53, 44], accompanied by the emergence of various video editing techniques [27, 49, 54, 26, 35, 10, 13, 23, 9, 58]. Among these techniques, video inpainting plays a crucial role by enabling the regeneration of content in corrupted or missing regions [48, 59, 2, 21, 50, 58, 20, 55, 56, 15, 25, 58, 24]. Generally, video inpainting tasks can be categorized into two main types: one focuses on generating or editing objects within specified regions with prompts, while the other aims to remove objects based on provided masks.

**Text-guided Video Inpainting.** For the first type of video inpainting, which involves generating or editing content within a masked region, significant progress has been made in recent years. VideoComposer [48] is the first diffusion model capable of performing text-guided video inpainting. It supports multiple conditioning inputs and can handle various tasks within a unified framework. AVID [55] proposed a method for text-guided video inpainting of arbitrary-length sequences using natural language prompts. COCOCO [59] improves consistency and controllability through damped global attention and enhanced cross-attention to text. VIVID [20] introduces a large-scale dataset with 10M image and video samples for localized editing, enabling training of a powerful text-guided inpainter. MTV-Inpaint [50] is capable of handling both traditional scene completion and novel object insertion. VideoPainter [2] performs inpainting using a DiT-based architecture, incorporating

an efficient context encoder to process masked inputs and injecting backbone-aware background information into a pre-trained video DiT for plug-and-play consistent video inpainting.

**Video Object Removal.** In contrast to text-guided inpainting, some methods focus specifically on removing objects from videos. ProPainter [56] uses optical flow to first complete the motion in masked regions, then synthesizes inpainted content accordingly. FFF-VDI [24] propagates noise latents from future frames to fill masked areas in the initial latent space, then finetunes a pre-trained image-to-video diffusion model for final synthesis. FloED [15] integrates both optical flow and text prompts, injecting embeddings from both modalities into the inpainting model for object removal. DiffuEraser [25] combines a flow-guided inpainting model with DDIM inversion for improved inpainting fidelity. Senorita-Remover, proposed in [58], introduces instruction-based object removal, using positive prompts to guide removal and negative prompts to suppress unintended object generation in masked areas.

**Remark.** Compared with previous methods, our **MiniMax-Remover** does not rely on additional prior or prompts. It offers fast inference speed with only 6 sampling steps, no CFG. Despite its simplicity, it achieves a higher object removal success rate and prevent hallucinated object and artifacts generation.

## 3   Preliminary

**Flow Matching.** Given an input $x \in \mathbb{R}^{f \times w \times h \times c}$, a pretrained VAE encoder maps it to a latent code $z_0 = \mathcal{E}(x_0) \in \mathbb{R}^{f_1 \times w_1 \times h_1 \times c_1}$. The model input is a noisy latent $z_t$, where $z_t = t\epsilon + (1-t)z_0$, where $\epsilon \sim \mathcal{N}(0, I)$ and $t \in [0,1]$. $t$ is usually sampled from a logit-normal distribution. The target velocity is $v = \epsilon - z_0$. The optimization process of Flow Matching is defined as,

$$\theta^* = \underset{\theta}{\operatorname{argmin}} \, \mathbb{E}_{t, z_t} \left[ \| u_\theta(z_t, t) - v \|_2^2 \right], \tag{1}$$

$u_\theta$ denotes the model parameterized by $\theta$. The model can be a DiT [38] or UNet [43] architecture.

**Diffusion-based Video Inpainting.** This approach is to reconstruct masked video regions using a conditional diffusion model. The denoising network $u_\theta$ takes a noisy latent $z_t$, a masked latent $z_m$, a latent code $\bar{m}$ for the mask $m$, and a condition $c$ as inputs. It optimizes the following objective,

$$\theta^* = \underset{\theta}{\operatorname{argmin}} \, \mathbb{E}_{t, z_t} \left[ \| u_\theta(z_t, z_m, \bar{m}, c, t) - \epsilon_t \|_2^2 \right]. \tag{2}$$

Generally, $c$ could be optical flow or text prompts.

**Classifier-Free Guidance (CFG).** CFG [18] improves conditional generation by training the model with and without condition $c$. The network jointly learns the following processes,

$$u_\theta(z_t, t, c) \propto -\nabla_{z_t} \log p_\theta(z_t | c), \qquad u_\theta(z_t, t, \varnothing) \propto -\nabla_{z_t} \log p_\theta(z_t), \tag{3}$$

In the inference stage, it combines the following two modes:

$$\hat{u}_\theta(z_t, t) = u_\theta(z_t, t, \varnothing) + w \cdot (u_\theta(z_t, t, c) - u_\theta(z_t, t, \varnothing)), \tag{4}$$

where $w$ controls the conditioning strength. However, employing CFG at inference time doubles the computational cost, as it requires two forward passes per sampling step.

**MiniMax Optimization.** It is widely employed in classical convex optimization [5], robustness enhancement and generative adversarial networks (GANs) [14]. The objective is defined as,

$$\theta^* = \underset{\theta}{\operatorname{argmin}} \, \underset{z}{\max} \, \mathcal{L}(f(z, \theta), y) \tag{5}$$

where $\mathcal{L}$ is the loss function, $f$ is the model with parameter $\theta$, $z$ and $y$ are the input and ground truth.

## 4   Methodology

### 4.1   Overall Framework

As shown in Figure 3, our method can be summarized as two stages.

***Stage 1: Training a lightweight video object removal model***. Our method follows the standard video inpainting pipeline, but with two simple yet effective improvements. First, we design a lightweight architecture by removing irrelevant components. Unlike many existing methods [59, 55, 48, 25], we do not use text prompts or extra inputs such as optical flow and text prompts, allowing us to remove all cross-attention layers. Second, we introduce two contrastive condition tokens to guide the inpainting process: a *positive token*, which encourages the model to fill in content within the masked regions, and a *negative token*, which discourages the model from generating unwanted objects in those areas. It should be noted that *unlike prior works* [55, 56, 24]*, we only use object mask and do not rely on additional conditions.*

***Stage 2: Enhancing model robustness and efficiency via human-guided minimax optimization.*** We first use the model in Stage 1 to generate inpainted video samples and then ask human annotators to identify successful results. Using this curated subset, we apply a minimax optimization training scheme to enhance the model's robustness and generation quality. Furthermore, the distilled remover can use as few as 6 steps without the help of CFG, resulting a fast inference. The resulting improved model is referred to as **MiniMax-Remover**.

## 4.2 Stage 1: A Simple Architecture for Video Object Removal

Our method is built on the pretrained video generation model Wan2.1-1.3B [44], which is a Flow Matching model with DiT architecture.

### 4.2.1 Model Architecture

**Input Layer.** We start by concatenating three types of latents: a noisy latent $z_t$, a masked latent $z_m$, and a mask latent $\bar{m}$. They are defined as $z_t = t\epsilon + (1-t)z_0$, where $z_0 = \mathcal{E}(x)$, $z_m = \mathcal{E}(m \odot x)$, and $\bar{m} = \mathcal{E}(m)$. Here, $x$ denotes the input video, $t \in [0, 1]$ is the diffusion timestep, and $\mathcal{E}$ is the VAE encoder. Each latent has 16 channels, resulting in a concatenated input of 48 channels. To accommodate this input, we modify the pretrained patch embedding layer to accept 48 channels instead of the original 16. Specifically, the first 16 channels retain the pretrained weights, while the remaining 32 channels are zero-initialized.

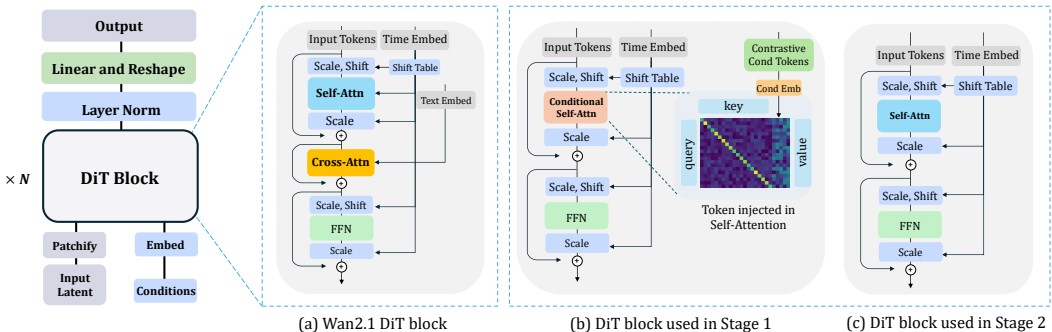

(a) Wan2.1 DiT block      (b) DiT block used in Stage 1      (c) DiT block used in Stage 2

Figure 2: The comparison between different blocks. (a) the original Wan2.1 DiT block; (b) DiT block with contrastive tokens (positive or negative token); (c) the block with removing the CFG.

**Removing Cross Attention from Pretrained DiT Block.** In the pretrained Wan2.1-1.3B [44] model, time information is injected via a *shift table*, a bias-based mechanism to encode timestep information. Additionally, the model employs a cross-attention module to incorporate textual conditioning during video generation. However, for the task of video object removal, textual input is often unnecessary or ambiguous. Therefore, in our model, we remove the textual cross-attention layers in the DiT blocks, while retaining the shift table to preserve time information.

**Injecting Contrastive Condition Tokens via Self-Attention.** To enable conditional inpainting, we introduce two learnable condition tokens, denoted as $c^+$ (positive token) and $c^-$ (negative token), as a replacement for text embeddings. We refer to these tokens as **contrastive condition tokens**.

Removing the cross-attention from DiT introduces a challenge: how to effectively inject conditional information without relying on textual prompts. A straightforward approach is to repurpose the shift

table to incorporate both timestep and condition information. However, our experiments show that this approach leads to unsatisfactory conditional inpainting results. To achieve more effective conditioning, we instead inject the contrastive condition tokens into the DiT block via the self-attention module. Specifically, we employ a learnable embedding layer to project the conditional token into a high-dimensional feature, and then split the feature into 6 tokens to increase control ability in attention computation process. These condition tokens are concatenated with the original keys and values in the self-attention module, enabling effective conditioning with minimal architectural modifications. For clarity, consider an example: in the original self-attention module, let $Q \in \mathcal{R}^{n \times d}$, $K \in \mathcal{R}^{n \times d}$, $V \in \mathcal{R}^{n \times d}$, after injecting the condition tokens, $Q \in \mathcal{R}^{n \times d}$, $K \in \mathcal{R}^{(n+6) \times d}$, $V \in \mathcal{R}^{(n+6) \times d}$. The updated DiT block after injecting contrastive condition tokens is illustrated in Figure 2(b).

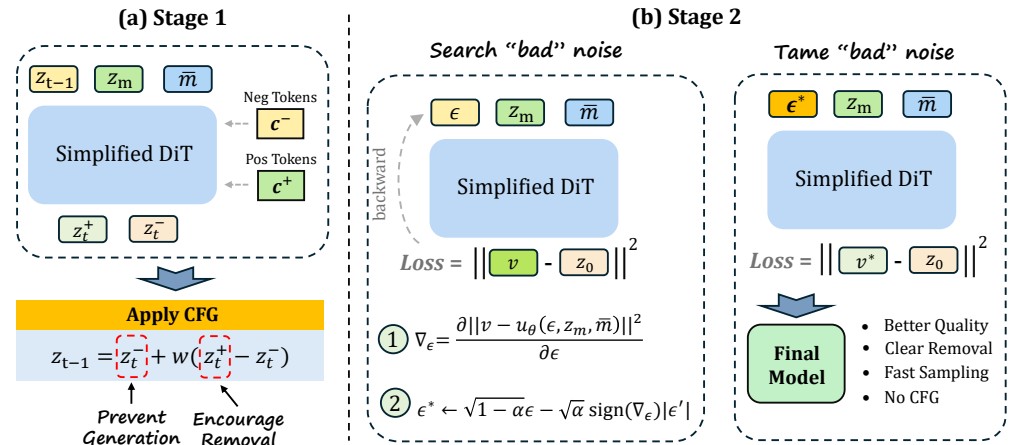

Figure 3: The pipeline of our two-stage method.

### 4.2.2 Contrastive Conditioning for Object Removal

We employ the positive condition token $c^+$ to guide the remover network in learning object removal and encourage the model to generate target objects under the guidance of $c^-$. Specifically, when applying classifier-free guidance, $c^+$ serves as the positive condition and $c^-$ as the negative condition, steering the model away from regenerating the target objects, preventing the reappearance of undesired objects within the masked regions. To train this behavior, we use two complementary strategies: **First, training the model to remove objects.** We randomly select masks from other videos and apply them to the current original video. The original video is used as the ground truth, and the model is conditioned on the positive prompt $c^+$. Since these masks usually don't match any real object in the current video, the model learns to fill the masked areas using information from the surroundings, instead of trying to recreate an object that fits the mask shape. This helps the model focus on inpainting the background rather than generating new objects. **Second, training the model to generate objects.** We use accurate masks that tightly cover real objects in the same video, along with the negative prompt $c^-$. This teaches the model to connect the shape of the mask with the object it should generate. During inference, we can use $c^-$ as a negative signal to prevent the model from reconstructing the object in those areas. More details are provided in the Appendix.

Given latent features $z_t$, masked latents $z_m$, mask latents $\bar{m}$, and timestep $t$, the network $u_\theta$ predicts:

$$\hat{z}_{t-1}^+ = u_\theta(z_t, z_m, \bar{m}, c^+, t), \qquad \hat{z}_{t-1}^- = u_\theta(z_t, z_m, \bar{m}, c^-, t). \tag{6}$$

Using CFG with guidance weight $w$, the final prediction is computed as:

$$\hat{z}_{t-1} = \hat{z}_{t-1}^- + w(\hat{z}_{t-1}^+ - \hat{z}_{t-1}^-). \tag{7}$$

The positive token guides the remover network in learning object removal, and the negative token encourages the model to generate object content. We would like to point out that we employ CFG during Stage 1 of training to facilitate conditional learning. However, CFG is removed in Stage 2 to improve inference efficiency.

### 4.2.3 Limitations of Stage 1

Despite improvements in simplicity and speed, the current model still faces three limitations. (1) CFG doubles the inference time and requires manual tuning of the guidance scale, which can vary across videos. (2) sampling 50 diffusion steps per frame remains time-consuming. (3) Artifacts or undesired object regeneration may occasionally occur within the intended object removal region, indicating that the contrastive signal is not yet fully effective. To address these limitations, we introduce our Stage 2 method, which is designed to enhance robustness, quality, and efficiency.

### 4.3 MiniMax-Remover: Distill a Stronger Video Object Remover from Human Feedback

Although our video object remover is trained with contrastive conditioning, it still produces noticeable artifacts and occasionally reconstructs the very object it is supposed to erase. Upon closer examination, we observe that these failure cases are strongly correlated with specific input noise pattens. This insight motivates our objective: to identify such 'bad noise" and train the object removal model to be robust against it.

The minmax optimization also allows us to escape from the CFG usage. In Stage 2, we eliminate CFG to improve sampling efficiency. Specifically, during training, we omit both the positive and the negative condition token. We choose to leave more analysis on this design in Appendix.

Moreover, performing inference with 50 steps is time-consuming. To address this, we distill the model in Stage 1 using the Rectified Flow method [31] to accelerate the sampling process. To build dataset for minimax optimization, we manually selected 10K video pairs from the 17K generated by the Stage 1 model as training data. This not only reduces the number of sampling steps but, with the help of min-max optimization, also encourages the model to produce better object removal results. Consequently, our model formulation changes from $u_\theta(z_t, z_m, \bar{m}, c, t)$ to $u_\theta(z_t, z_m, \bar{m}, t)$. We formulate our stage-2 training as a minimax optimization problem:

$$\min_{\theta} \max_{\epsilon} \mathbb{E}_t ||u_\theta(z_t, z_m, \bar{m}, t) - v||^2, \quad \text{where } v = \epsilon - z_{\text{succ}} \text{ and } z_t = t\epsilon + (1-t)z_0. \quad (8)$$

Therefore, the inner maximization seeks bad noise that maximizes the prediction error, effectively finding challenging input noise. The outer minimization then updates the model parameters $\theta$ to be robust against such adversarial noise. This minimax optimization strategy encourages the model to remain stable even under difficult or misleading input noise.

#### 4.3.1 Search for "Bad" Noise

One of the key challenges in Equation 8 is how to effectively identify a "bad" noise sample $\epsilon$. Rather than directly maximizing the loss in Equation 8 using successful object removal cases, we instead minimize the loss with respect to a bad target: specifically, the original video, which fails to meet the objective of object removal. This leads to the following reformulated objective:

$$\min_{\epsilon \sim \mathcal{N}(0,I)} \left\| u_\theta(z_t = \epsilon, z_m, \bar{m}, t = 1.0) - v \right\|^2, \quad \text{where } v = \text{sg}(\epsilon) - z_{\text{org}}, \quad (9)$$

where $\text{sg}(\cdot)$ is the stop gradient operation and $z_{\text{org}}$ denotes the latent code of the original video. It is important to note that, in Equation 9, we omit the expectation $\mathbb{E}_{t,z_t}$, because we fix the diffusion timestep to $t = 1.0$, making $z_t = \epsilon$.

Starting from a random noise $z_{t=1} = \epsilon$, we compute the gradient of the loss function with respect to $\epsilon$ via backpropagation. This gradient is given by:

$$\nabla_\epsilon = \partial \|u_\theta(\epsilon, z_m, \bar{m}, t = 1) - v\|^2 / \partial \epsilon \quad (10)$$

After obtaining the gradient with respect to the noise, we can construct a new adversarial ("bad") noise sample $\epsilon^*$ as follows,

$$\epsilon^* \leftarrow \sqrt{1-\alpha}\,\epsilon - \sqrt{\alpha}\,\text{sign}(\nabla_\epsilon) \cdot |\epsilon'|, \quad (11)$$

where $\epsilon'$ is a newly sampled noise, $\text{sign}(\nabla_\epsilon)$ is the sign of the gradient obtained in Equation 10, and $\alpha \in [0, 1]$ is a randomly sampled scalar. We use only the sign of the gradient to suppress the influence of gradient magnitude, ensuring more stable updates. This resulting noise $\epsilon^*$ encodes object-related information that tends to reconstruct the original content, thereby serving as a challenging adversarial noise. Meanwhile, the formulation in Equation 11 preserves the approximate distributional properties of standard Gaussian noise, making $\epsilon^*$ compatible with the diffusion process.

### 4.3.2 Optimize for Robustness to "Bad" Noise

In Stage 2, we enhance the robustness of the model by fine-tuning it on adversarial noise samples. We minimize the following objective,

$$\min_{\boldsymbol{\theta}} \mathbb{E}_{t,\boldsymbol{z}_t^*} \left\| \boldsymbol{u}_{\boldsymbol{\theta}}(\boldsymbol{z}_t^*, \boldsymbol{z}_m, \bar{\boldsymbol{m}}, t) - \boldsymbol{v}^* \right\|^2, \quad \text{where } \boldsymbol{v}^* = \boldsymbol{\epsilon}^* - \boldsymbol{z}_{\text{succ}} \text{ and } \boldsymbol{z}_t^* = t\boldsymbol{\epsilon}^* + (1-t)\boldsymbol{z}_{\text{succ}}, \quad (12)$$

$\boldsymbol{z}_{\text{succ}}$ is the latent for a successful inpainted video. In Equation 12, we sample a timestep $t$, and construct a noisy latent input $\boldsymbol{z}_t^*$ according to the previously generated $\boldsymbol{\epsilon}^*$ and $\boldsymbol{z}_{\text{succ}}$. In practice, we train on a mixture of data. Specifically, one-third of training samples are drawn from our curated 10K set with their associated adversarial noises, while the remaining two-thirds are standard WebVid-10M videos with randomly generated object masks. This mixed strategy ensures the model remains effective on both clean and challenging inputs, ultimately leading to improved generalization and resilience against failure cases. We refer to the model after Stage 2 training as MiniMax-Remover.

### 4.3.3 Advantages of MiniMax-Remover

MiniMax-Remover owns several key advantages:

- **Low Training Overhead**. It only back-propagates once to search the "bad" noise, and trains a remover with simplified architectures which reduces the memory consumption.

- **Fast Inference Speed**. MiniMax-Remover only uses as few as 6 sampling steps without CFG, resulting in significantly faster inference compared to prior methods.

- **High Quality**. Since the model is trained to be robust against "bad" noises, it rarely produces unexpected objects or visual artifacts in the masked regions, resulting in a higher quality.

## 5 Experiments

**Training Dataset.** In Stage 1, we use Grounded-SAM2 [29, 42] and captions from CogVLM2 [19] to generate masks on the watermark-free WebVid-10M dataset [1]. Approximately 2.5M video-mask pairs are randomly selected for training. In Stage 2, we collect 17K videos from Pexels [39] and apply the same annotation process as in Stage 1. These are further processed using the model from Stage 1, and 10K videos are manually selected for Stage 2 training.

**Training Details.** For Stage 1, we initialize our model with Wan2.1-1.3B [44]. Newly added layers, such as the embedding layer, are randomly initialized. The first 16 channels of the patch embedder are copied from Wan2.1, while the remaining 32 are zero-initialized. Training uses a batch size of 128, input frame length of 81, and resolutions randomly sampled from $336 \times 592$ to $720 \times 1280$. We set the first N mask frames to 0 to support the any-length inpainting by applying sliding windows, using a random ratio of 0.1. We use AdamW optimizer [32] with a constant learning rate of $1e-5$, weight decay of $1e-4$, and train for 10K steps. In Stage 2, we reuse the model from Stage 1, excluding the embedding layer since no external conditions are needed. One-third of the training iterations apply min-max optimization; the rest follow standard training using unrelated masks from WebVid [1]. Hyperparameters remain the same as in Stage 1. All experiments are conducted on 8 A800 GPUs (80GB each) and take about two days in total.

**Inference Details.** We perform inference using RTX 4090 GPUs. With an input resolution of 480p and a frame length of 81, *the inference takes approximately 24 seconds per video* and consumes around 14GB peak GPU memory(DiT for 8GB, VAE decoding for 6GB), using 6 sampling steps.

**Baselines.** We compare our method with ProPainter [56], VideoComposer [48], COCOCO [59], FloED [15], DiffuEraser [25], VideoPainter [2] and VACE [21]. We set the evaluate frame length as 32. To evaluate with same frame length, we expand input frame length for VideoComposer [48] and FloED [15]. The rest video inpainters are used their default frame length of their code bases. The frame resolutions are used with their default resolutions.

**Metrics.** We evaluate background preservation using SSIM and PSNR. TC evaluates the temporal consistency, follows COCOCO [59] and AVID [55] with CLIP-ViT-h-b14 [41] to compute features. GPT-O3 [36] serve as objective metrics. We evaluate these metrics on DAVIS datasets and 200 randomly selected Pexels videos to show generalizations across different datasets. Note that the

200 Pexels videos are not contained in our training datasets, and masks are extracted by Grounded-SAM2. In the user study, participants are presented with a multiple-choice questionnaire to identify which video most effectively removed the target objects from the original video, without introducing blurring, visual artifacts, or hallucinated content within the masked areas.

Table 2: Comparison of different methods on the DAVIS [40] and Pexels [39] datasets. The best results are highlighted in **bold**. "TC" denotes temporal consistency, "VQ" stands for visual quality, and "Succ" represents the success rate.

| Method | DAVIS Dataset | | | | | | Pexels Dataset | | | | | |
|---|---|---|---|---|---|---|---|---|---|---|---|---|
| | Quantitative Results | | | GPT-O3 Eval | | User Preference | Quantitative Results | | | GPT-O3 Eval | | User Preference |
| | SSIM | PSNR | TC | VQ | Succ | | SSIM | PSNR | TC | VQ | Succ | |
| Propainter [56] | 0.9748 | 35.33 | 0.9769 | 5.68 | 56.67 | 34.55% | 0.9746 | 35.76 | 0.9813 | 5.07 | 35.5 | 23.28% |
| VideoComp [48] | 0.8689 | 30.66 | 0.9529 | 2.12 | 7.78 | 0.25% | 0.8865 | 30.26 | 0.9573 | 3.10 | 14.5 | 0.245% |
| COCOCO [59] | 0.8863 | 32.10 | 0.9511 | 3.40 | 12.22 | 1.96% | 0.9145 | 30.84 | 0.9693 | 4.32 | 30.0 | 0.245% |
| FloED [15] | 0.9053 | 32.02 | 0.9630 | 5.13 | 45.56 | 10.54% | 0.9350 | 34.82 | 0.9688 | 4.19 | 24.5 | 12.01% |
| DiffuEraser [25] | 0.9818 | 34.42 | 0.9767 | 5.71 | 56.67 | 42.40% | 0.9859 | 34.41 | 0.9800 | 5.89 | 59.0 | 20.83% |
| VideoPainter [2] | 0.9654 | 34.60 | 0.9620 | 3.80 | 17.98 | 1.47% | 0.9821 | 36.49 | 0.9847 | 5.68 | 48.0 | 7.11% |
| VACE [21] | 0.9102 | 31.92 | 0.9747 | 3.12 | 8.89 | 2.21% | 0.9102 | 32.33 | 0.9898 | 6.27 | 49.5 | 23.77% |
| Ours (6 steps) | 0.9842 | 36.56 | 0.9770 | 6.26 | 82.22 | 58.08% | 0.9873 | 36.98 | 0.9905 | 6.87 | 76.5 | 62.01% |
| Ours (12 steps) | 0.9846 | 36.62 | 0.9772 | 6.36 | 84.44 | 63.24% | 0.9872 | **37.02** | **0.9906** | 6.86 | 80.5 | **67.15%** |
| Ours (50 steps) | **0.9847** | **36.66** | **0.9776** | **6.48** | **91.11** | **64.22%** | **0.9878** | 36.98 | 0.9905 | **6.90** | **81.0** | 63.97% |

## 5.1 Quantitative Comparison

As show in Table 2, our method outperforms previous baselines on all 90 DAVIS videos, achieving an SSIM of 0.9847 and a PSNR of 36.66. Notably, even with only 6 sampling steps, our approach can generate high-quality videos while effectively preserving background details. Furthermore, our method exhibits superior temporal consistency, significantly outperforming generative models such as VACE [21], and even surpassing the traditional inpainting method Propainter [56]. These results demonstrate that our model consistently produces visually pleasing and high-quality video object removal. A similar trend is observed on 200 Pexels videos, where our method achieves the highest SSIM, PSNR, and temporal consistency scores. Moreover, reducing the number of sampling steps does not significantly degrade the removal performance.

| Org Video | VideoComposer | COCOCO | FloED | Ours-6 Steps |
|---|---|---|---|---|
| Propainter | Diffueraser | VideoPainter | VACE | Ours-50 Steps |
| Org Video | VideoComposer | COCOCO | FloED | Ours-6 Steps |
| Propainter | Diffueraser | VideoPainter | VACE | Ours-50 Steps |

Figure 4: The visual results of our object remover. The video on the left depicts the original video, while the video on the right displays the edited videos. *Best viewed with Acrobat Reader. Click the images to play the animation clips.*

## 5.2 Qualitative Results

To assess the visual quality and object removal success rate, we utilize GPT-O3 [36], a powerful reasoning large language model, by querying it with evaluation prompts. The quality score ranges from 1 (worst) to 10 (best). According to GPT-O3 evaluation, our method achieves a higher score of 6.48, compared to 5.71 from the best previous method, indicating clearer and more visually appealing removal results. Regarding removal success rate, we prompt the GPT-O3 to determine whether the target objects were effectively removed. Our method achieves a remarkable 91.11% success rate on the DAVIS dataset, far surpassing the previous best of 56.67%. On the Pexels dataset, our method also outperforms previous state-of-the-art approaches, with an 81% success rate compared to 59.0%. Additionally, our method achieves a higher score 6.90, versus 6.27 from prior best methods. For the user preference, it has similar trends, our method achieves the best score on both two datasets compared with previous best remover, 64% vs 42.40% and 67.15% vs 23.77%, respectively.

Table 3: Ablation Study for two stages' training. The best results are **boldfaced**.

| Method | Stage | Structure | Condition | •Quantitative Results | | | •GPT-O3 Evaluation | |
|--------|-------|-----------|-----------|------|------|------|----------------|--------------|
| | | | | SSIM | PSNR | TC | Visual Quality | Success Rate |
| Ab-1 | Stage 1 | ShiftTable+Cross-Attn | Prompt | 0.9737 | 34.77 | 0.9756 | 6.27 | 51.11 |
| Ab-2 | Stage 1 | ShiftTable+Cross-Attn | Tokens | 0.9747 | 35.09 | 0.9752 | 6.37 | 56.67 |
| Ab-3 | Stage 1 | ShiftTable | Tokens | 0.9682 | 34.87 | 0.9743 | 6.42 | 53.33 |
| Ab-4 | Stage 1 | ShiftTable+Self-Attn | Tokens | **0.9798** | **35.87** | **0.9773** | **6.39** | **71.11** |

| Method | Stage | Data Type | Input Noise | •Quantitative Results | | | •GPT-O3 Evaluation | |
|--------|-------|-----------|-------------|------|------|------|----------------|--------------|
| | | | | SSIM | PSNR | TC | Visual Quality | Success Rate |
| Ab-1 | Stage 2 | WebVid Data | Random Noise | 0.9781 | 35.49 | 0.9759 | 6.27 | 65.56 |
| Ab-2 | Stage 2 | Human Data | Random Noise | 0.9796 | 35.21 | 0.9772 | 6.36 | 72.22 |
| Ab-3 | Stage 2 | Human Data | Bad Noise-Adv | **0.9847** | **36.66** | **0.9776** | **6.48** | **91.11** |

## 5.3 Ablation Study

To understand the impact of each component and modification in our method, we conduct a step-by-step ablation study. All experiments use 50 sampling steps.

**Stage 1.** We begin by examining the role of the text encoder and prompt-based conditioning. In the comparison between Ab-1 and Ab-2 (Table 3), we replace the text encoder and prompts with learnable contrastive tokens. The results show no significant drop in performance, indicating that the text encoder is redundant for the removal task when suitable learnable tokens are used instead. Next, comparing Ab-2 and Ab-3, we observe a slight performance degradation after removing the cross-attention module from the DiT. However, when we introduce learnable contrastive condition tokens into the self-attention layers (Ab-4), the results not only recovers but also surpass that of Ab-1. This demonstrates the effectiveness of our simplified DiT architecture.

**Stage 2.** We compare models trained with and without human-annotated data. The results (Ab-1 vs. Ab-2) show that using manually labeled data alone does not significantly improve performance, likely due to the limited size (10K videos) and diversity of the dataset, which hinders generalization. Furthermore, we compare different noise types used during training (Ab-2 to Ab-3). We find that adding "bad noise" (artificially degraded inputs) into training helps improve performance significantly.

# 6 Conclusion

In this paper, we propose *MiniMax Remover*, a two-stage framework for object removal in videos. In Stage 1, we simplify the pretrained DiT by removing cross-attention and replacing prompt embeddings with contrastive condition tokens. In Stage 2, we apply min-max optimization: the *max* step searches for challenging noise inputs that lead to failure cases, while the *min* step trains the model to successfully reconstruct the target from these adversarial inputs. Through this two-stage training, our method achieves cleaner and more visually pleasing removal results. Since it requires no classifier-free guidance (CFG) and uses only 6 sampling steps, inference is significantly faster. Extensive experiments demonstrate that our model delivers impressive removal performance across multiple benchmarks.

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

# 7 More Details of Stage-1 Training

Here, we provide further details on the training process of Stage 1. We adopt the same mask selection strategy as in Senorita-Remover [58]. In this paper, we propose to simplify the pretrained DiT architecture and replace the prompts with learnable contrastive condition tokens.

The most challenging issue in object removal is the undesired regeneration problem, where the model tends to regenerate objects in the masked region that share a similar shape to the mask itself. To address this issue, as illustrated in Figure 5, we introduce contrastive condition tokens: a positive token $c^+$ to guide the model towards object removal, and a negative token $c^-$ to guide object generation within the masked region. This allows the model to leverage CFG for effective object removal while preventing the reappearance of similar objects in the masked areas.

In Figure 5(a), when selecting masks, we randomly select masks from other unrelated videos, and paste the masks into the input video. These masks are designed to differ in shape from the actual objects within the masked region of the input video. As a result, the model, trained on such data, tends to generate content that differs from the mask shape during inference. This process is considered as the positive condition $c^+$. The position condition learns to remove the masked objects.

In Figure 5(b), we incorporate training samples using their corresponding precise masks. In these cases, the masked region contains an object whose shape closely matches the mask. Training on such data encourages the model to regenerate similar objects in the masked region when conditioned on $c^-$. This contrastive training strategy helps the model distinguish between removal and generation tasks more effectively. This process is considered as the negative condition $c^-$. The negative condition learns to generate the masked objects.

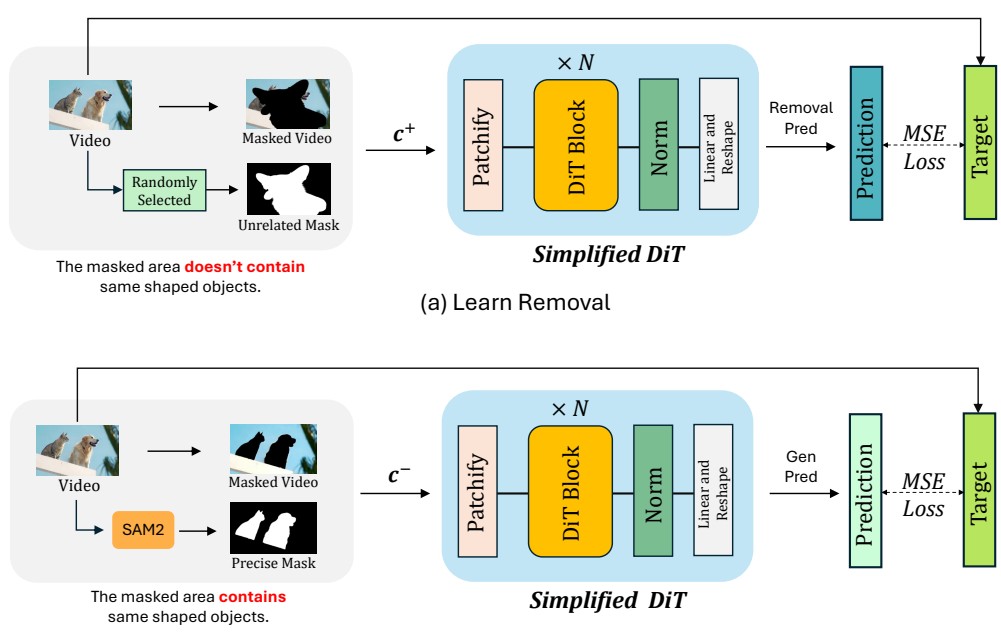

Figure 5: Training framework of the Stage-1. (a) denotes the positive condition process, and the position condition learns to remove the masked objects. and (b) represents the negative process, and the negative condition learns to generate the masked objects..

# 8 Explanation Of Why CFG Can Be Discarded

## 8.1 MiniMax Optimization

In our work, we formulate the min-max remover using the following objective:

$$\min_{\boldsymbol{\theta}} \max_{\boldsymbol{\epsilon}^*} \|\boldsymbol{u}_{\boldsymbol{\theta}}(\boldsymbol{\epsilon}^*, \boldsymbol{z}_m, \bar{\boldsymbol{m}}) - \boldsymbol{z}^*)\|^2 , \tag{13}$$

where $u$ is a DiT network parameterized by $\theta$, and $v^* = \epsilon^* - z_{\text{succ}}$, with $z_{\text{succ}} = \mathcal{E}(x_{\text{succ}})$. Here, $x_{\text{succ}}$ is a video clip filtered and accepted by human annotators, and $\mathcal{E}$ is the autoencoder.

In practice, to search for the "bad" noise, we replace the inner maximization with:

$$\min_{\epsilon^*} \|u_\theta(\epsilon^*, z_m, \bar{m}) - v^*\|^2, \tag{14}$$

where $v^* = \epsilon^* - z_{\text{org}}$, and $z_{\text{org}}$ corresponds to an undesired generation. This approach finds a noise $\epsilon^*$ that causes $u_\theta$ to fail. By subsequently training $\theta$ to minimize the objective even under such challenging noise, the model learns to produce outputs that deviate from the poor generation target $v^*$.

## 8.2 Minimax Optimization Encourages Correct Removal and Discourages "Bad" Removal

**Minimax Optimization Encourages Correct Removal**. Based on the principles of minimax optimization discussed above, we have that:

$$||u_\theta(\epsilon, z_m, \bar{m}) - (\epsilon - z_{\text{succ}})||^2 \leq ||u_\theta(\epsilon^*, z_m, \bar{m}) - (\epsilon^* - z_{\text{succ}})||^2. \tag{15}$$

This inequality implies that minimizing the loss on the "bad" input noise $\epsilon^*$ leads to improved performance on the clean input noise $\epsilon$. Consequently, the model is incentivized to generate outputs that increasingly align with desired removals.

**Minimax Optimization Discourages "Bad" Removals**. As shown in the formulation,

$$\min_{\theta} \|u_\theta(\epsilon^*, z_m, \bar{m}) - (\epsilon^* - z_{\text{succ}})\|^2,$$

the model is optimized to ensure that, upon convergence, no "bad" input noise $\epsilon^*$ can lead to a failure-inducing removal. Consequently, the model inherently avoids generating such "bad" removals.

In summary, this min-max training strategy enables the model to avoid producing poor outputs (aligned with the Stage 1 negative condition $c^-$) while encouraging alignment with high-quality generations (aligned with the Stage 1 positive condition $c^+$). Therefore, after the minimax optimization, we can discard the classifier-free guidance (CFG), and the quality of removal results remains unaffected.

**Experiments**. Here, we conduct experiments to demonstrate that discarding CFG does not significantly affect performance. To ensure a fair comparison, we use the human-annotated data from Stage 2 and reuse the Stage 1 model without removing the tokens injected into the self-attention layer. Thus, the ablation variant *ab-1* applies CFG by introducing two learnable contrastive tokens, $c^+$ and $c^-$. The sampling process consists of 50 steps, while all other training and inference settings remain consistent with those described in the main text. As shown in Table 4, using CFG leads to a slight improvement in background preservation and temporal consistency, but results in marginal decreases in visual quality and success rate. These results further support that the model performs robustly even without CFG.

Table 4: Comparison of results with/without CFG on the DAVIS [40] dataset. TC indicates Temporal Consistency, while Succ Rate stands for Success Rate. The best results are **boldfaced**.

| Method | •Quantitative Results | | | •GPT-O3 Evaluation | |
|---|---|---|---|---|---|
| | SSIM | PSNR | TC | Quality | Succ Rate |
| MiniMax-Remover(w CFG) | **0.9853** | **36.76** | **0.9778** | 6.45 | 90.00 |
| MiniMax-Remover(w/o CFG) | 0.9847 | 36.66 | 0.9776 | **6.48** | **91.11** |

# 9 Why Use Adversarial-Based Noise?

## 9.1 Comparison of Adversarial-Based Noise with Other Types of Noise

It might seem intuitive to replace adversarial-based noise with either random or inversion-based noise. However, our experiments clearly justify the use of adversarial noise.

Table 5: Ablation study on different noise types using the DAVIS dataset [40]. TC denotes Temporal Consistency, and Succ Rate indicates Success Rate. The best results are highlighted in **bold**.

| Method | Input Noise | •Quantitative Metrics | | | •GPT-O3 Evaluation | |
|---|---|---|---|---|---|---|
| | | SSIM | PSNR | TC | Quality | Succ Rate |
| Ablation Study-1 | Random Noise | 0.9796 | 35.21 | 0.9772 | 6.36 | 72.22 |
| Ablation Study-2 | Inversion-Based Noise | 0.9797 | 35.80 | 0.9768 | 5.81 | 70.00 |
| Ablation Study-3 (Ours) | Adversarial-Based Noise | **0.9847** | **36.66** | **0.9776** | **6.48** | **91.11** |

**Random noise** fails to guide the model towards adversarial directions. As a result, when training on limited data for a large number of iterations, it often leads to overfitting without improving model robustness or performance.

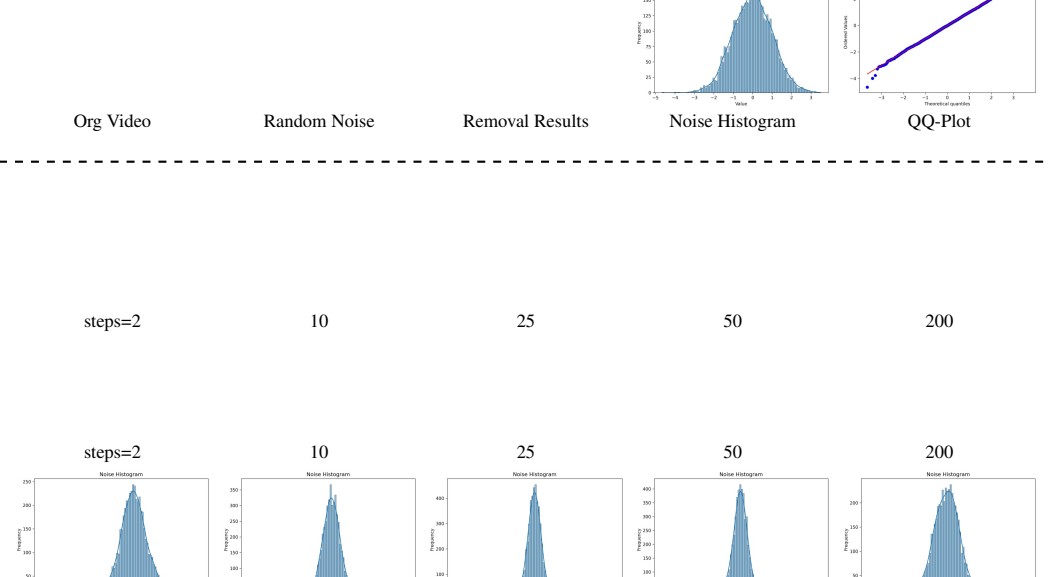

Figure 6: Visual results of the Stage-1 Object Remover under inversion-based noises. The top row presents the original video, random noise, removal results, the corresponding histogram, and QQ plot. Rows 2 to 5 at the bottom display the removal results with inversion-based noise, the inversion-based noise itself, and its associated histogram and QQ plot, respectively. *Best viewed with Acrobat Reader. Click the images to play the animation clips.*

**Inversion-based noise**, on the other hand, typically requires many optimization steps to reconstruct noise from latent representations. To keep the computational cost comparable to adversarial noise, we restrict this process to just three steps. However, such shallow inversion retains a significant amount of information from the latents, violating the assumption that noise follows a standard Gaussian distribution $\mathcal{N}(\mathbf{0}, \mathbf{I})$. Consequently, it cannot be considered "random" noise in a true sense.

In contrast, **adversarial noise** is constructed by updating the input noise in a direction that maximizes model failure, while still keeping it within the $\mathcal{N}(\mathbf{0}, \mathbf{I})$ distribution. Specifically, we apply the update

rule:

$$\epsilon^* \leftarrow \sqrt{1 - \alpha}\, \epsilon - \sqrt{\alpha} \cdot \text{sign}(\nabla_\epsilon) \cdot |\epsilon'|$$

This allows us to craft noise that remains statistically valid but strategically challenging for the model.

As shown in Table 5, adversarial-based noise significantly outperforms both random and inversion-based noise across all evaluation metrics, demonstrating its effectiveness in improving model robustness and generalization.

## 9.2 Determining Whether the Noise is Gaussian

To assess whether the noise follows a Gaussian distribution, we typically use visual tools such as histograms and QQ-plots. The histogram provides an overview of the noise distribution, while the QQ-plot compares the quantiles of the observed noise with those of a standard Gaussian distribution. If the points in the QQ-plot lie approximately along a straight line, this suggests that the noise is likely Gaussian.

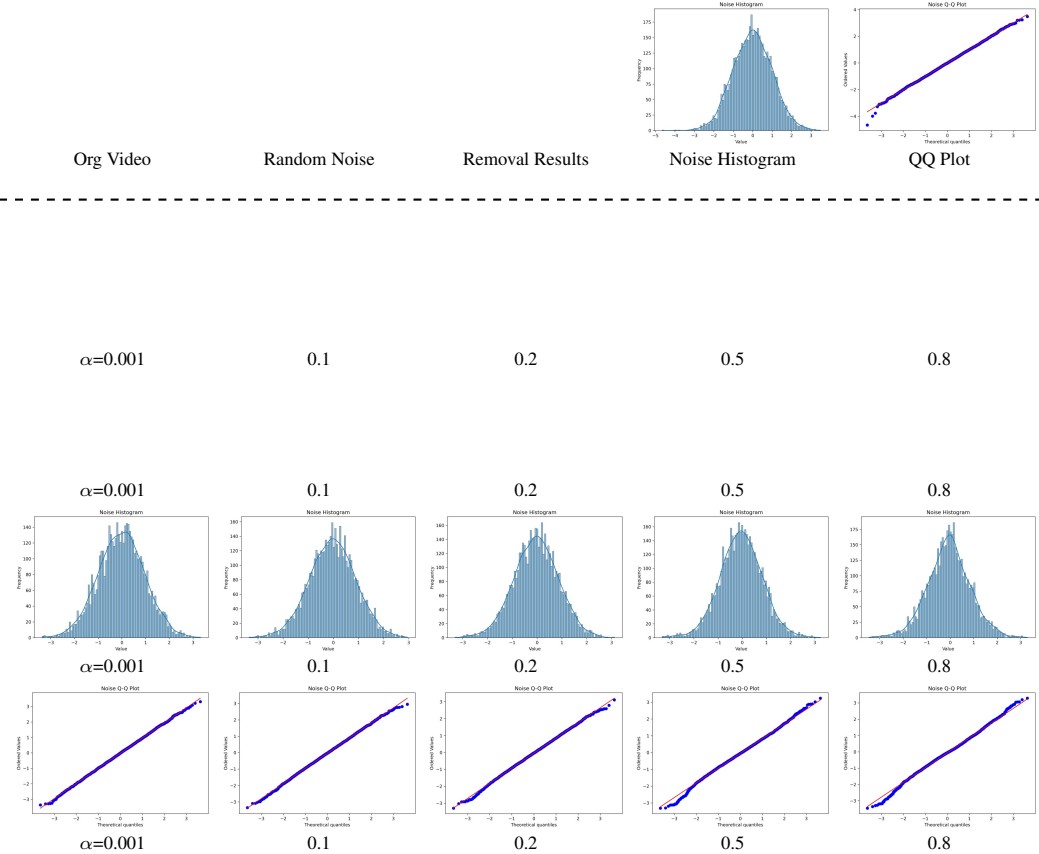

Figure 7: Visual results of the Stage-1 Object Remover under adversarial-based noises. The top row illustrates the original video, random noise input, the resulting object removal output, along with the corresponding noise histogram and QQ plot. The lower section comprises the object removal results under adversarial-based noise (second row), the visualization of the adversarial noise (third row), and its associated histogram and QQ plot (fourth row). *Best viewed with Acrobat Reader. Click the images to play the animation clips.*

**Inversion-Based Noise is Non-Gaussian**. As shown in Figure 6, both the histogram and QQ-plot reveal a clear deviation from the characteristics of Gaussian noise. Inversion-based noise exhibits a distribution that significantly differs from that of random noise. These visualizations support the conclusion that inversion-based noise does not follow a Gaussian distribution and is therefore unsuitable for training purposes. Moreover, the visualizations suggest that inversion-based noise retains information from the original latent representations, further compromising its utility as pure

noise. In addition, generating such noise requires multiple iterations of the inversion process, making it computationally expensive and impractical for large-scale applications.

**Adversarial-Based Noise Approximates a Gaussian Distribution**. Figure 7 presents the histogram and QQ-plot for adversarial-based noise. The visual evidence indicates that this type of noise closely approximates a Gaussian distribution. The visualizations also suggest that the noise is random and does not retain any information from the original latent representations. Both the histogram and QQ-plot are highly similar to those of standard Gaussian random noise. As a result, adversarial-based noise not only effectively perturbs model predictions but also serves as a suitable and reliable noise source for training. Notably, it can be generated efficiently using a single step of backpropagation.

| $\alpha = 0.001$ | 0.1 | 0.2 | 0.5 | 0.8 |

Figure 8: Visual Results of the Minimax-Remover with Adversarial-Based Noise. Adversarial-based noise is integrated into the Minimax-Remover, yielding visually robust and consistent removal performance. *Best viewed with Acrobat Reader. Click the images to play the animation clips.*

## 9.3 Testing Adversarial-Based Noise on the Robust MiniMax-Remover

We assess the performance of the robust Minimax-Remover when subjected to adversarial-based noise. Specifically, we apply adversarial noise to Minimax-Remover and gradually increase the noise level $\alpha$ from 0.001 to 0.8. As illustrated in Figure 8, our method successfully removes objects from video frames effectively, maintaining high visual quality when $\alpha < 0.5$. Even as the noise level increases beyond 0.5, the removal results may become slightly blurred, but no regenerated objects or visual artifacts are observed. This demonstrates that the MiniMax-Remover is resilient to high levels of adversarial-based noise and can still produce clean and reliable removal results under challenging conditions.

## 10 The Prompt used for GPT-O3 Evaluation

The evaluation is conducted using the `GPT-O3-2025-04-16` model. We employ the following prompt to assess visual quality and success rate.

> **Prompt for Quality Evaluation**
>
> **Human:** Please evaluate the visual quality of the object-removal result in this image pair: the left panel shows the original frame with the target object highlighted by a blue mask, and the right panel shows the frame after the object has been removed. Rate the quality of the removal on a scale from 1 to 10, where 10 means a seamless, high-resolution result with no visible artifacts, 5 indicates some noise or artifacts but overall acceptable quality, and 1 corresponds to an unacceptable, very poor result. Return only the numeric score.
>
> <Image> Image 1</Image>
> <Image> Image 2</Image>
> ...
> **GPT-O3:** ...

    1. The object has been completely removed.

    2. There is no visible blurriness in the removal area (not background blur, but unnatural foreground blur that is inconsistent with the surrounding context).

    3. There are no moiré patterns or artifacts in the region (e.g., unnatural textures that differ significantly from human visual expectations).

    4. No new, unwanted objects have been generated in the area (e.g., the tiger is removed but a bear appears instead, or another tiger is generated).

    5. Shadows not in the masked region are considered successfully removed.

Think it step by step. Then provide a final judgment by answering with "yes" or "no"

<Image> Image 1</Image>
<Image> Image 2</Image>

**GPT-O3:** ...

## 11 More Experimental Results

### 11.1 Evaluation of ReMOVE Metrics

Table 6: Comparison on the ReMOVE benchmark (higher is better).

| Method | ReMOVE (w/o Crop) | ReMOVE (w/ Crop) |
|---|---|---|
| ProPainter [56] | 0.8634 | 0.9259 |
| FloED [15] | 0.6950 | 0.8103 |
| DiffuEraser [25] | 0.8621 | 0.9212 |
| COCOCO [59] | 0.8009 | 0.8994 |
| VideoComposer [48] | 0.6605 | 0.7429 |
| VideoPainter [2] | 0.6955 | 0.8279 |
| VACE [21] | 0.5731 | 0.7723 |
| MiniMax-Remover (6 steps) | 0.8831 | 0.9401 |
| MiniMax-Remover (12 steps) | 0.8843 | 0.9405 |
| MiniMax-Remover (50 steps) | **0.8850** | **0.9416** |

Table 6 reports quantitative comparisons on the ReMOVE benchmark [7], a reference-free metric for evaluating object removal quality (higher is better). Our method consistently achieves the best performance under both ReMOVE (w/o Crop) and ReMOVE (w/ Crop) settings. Specifically, our 50-step model attains scores of 0.8850 and 0.9416, outperforming the strongest baseline ProPainter by +0.0216 and +0.0157, respectively. These results demonstrate the superior removal fidelity and spatial consistency of our approach.

Across all methods, the w/ Crop results are higher than the w/o Crop results, indicating that localized evaluation around the target region is generally easier. Nevertheless, our model maintains the highest accuracy in both configurations, highlighting its robustness to complex backgrounds and global contextual variations.

Overall, our model demonstrates strong generalization and stability across different evaluation conditions, surpassing prior state-of-the-art methods such as ProPainter [56], DiffuEraser [25], and COCOCO [59] on both global and localized ReMOVE metrics.

## 11.2 Can GPT-O3 be trusted?

Table 7: Consistency with human annotations on 100 removal samples.

| Assessment Type | GPT-4O | GPT-4O-mini | GPT-4-turbo | GPT-O3 |
|---|---|---|---|---|
| Acc in Failure Cases | 94.0% | 98.0% | 86.0% | **96.0%** |
| Acc in Success Cases | 88.0% | 26.0% | 58.0% | **94.0%** |
| Overall Accuracy | 91.0% | 62.0% | 72.0% | **95.0%** |

Table 8: Consistency with human annotations on 100 multi-object videos.

| Assessment Type | GPT-4O | GPT-4O-mini | GPT-4-turbo | GPT-O3 |
|---|---|---|---|---|
| Acc in Failure Cases | 90.0% | 92.0% | 86.0% | **98.0%** |
| Acc in Success Cases | **90.0%** | 48.0% | 48.0% | 86.0% |
| Overall Accuracy | 90.0% | 70.0% | 67.0% | **92.0%** |

We evaluate four models on two datasets with balanced splits of successes and failures and use masks to highlight the removed regions for evaluation. As reported in Table 7, GPT-O3 attains 96.0% accuracy in failure cases and 94.0% in success cases for 95.0% overall, GPT-4O records 94.0% and 88.0% for 91.0% overall, GPT-4-turbo reaches 86.0% and 58.0% for 72.0% overall, and GPT-4O-mini achieves 98.0% and 26.0% for 62.0% overall. Table 8 summarizes the 100 multi object video removals where GPT-O3 again leads with 98.0% in failure cases and 86.0% in success cases for 92.0% overall, while GPT-4O achieves 90.0% in both overall and success accuracy and GPT-4-turbo and GPT-4O-mini reach 67.0% and 70.0% overall respectively. Together Tables 7 and 8 show that GPT-O3 aligns best with human annotations by balancing recognition of failures and true successes.

## 11.3 Results on Multi-Object Videos

We evaluate on a multi-mask subset of DAVIS to stress multi-object editing and report SSIM, PSNR, Temporal Consistency, Visual Quality, and Success Rate in Table 9. MiniMax-Remover achieves the best overall results, with the 50-step variant reaching SSIM 0.9847, PSNR 36.69, TC 0.9780, VQ 6.70, and Succ 92.59%. This clearly outperforms strong baselines such as ProPainter at SSIM 0.9778 and PSNR 35.64 and DiffuEraser at SSIM 0.9769 and PSNR 35.08, while also yielding much higher VQ and task success. Traditional inpainting methods show reasonable pixel metrics but struggle in complex multi-mask interactions, which leads to lower VQ and Succ, as seen with VideoComposer at VQ 2.13 and Succ 9.26%. Performance scales with sampling steps for our method: 6 to 12 steps provides a notable gain, and 12 to 50 steps offers smaller but consistent improvements that help on hard cases. Overall, MiniMax-Remover balances temporal stability and detail restoration most effectively on DAVIS multi-mask videos.

## 11.4 Results on Dynamic Background Videos

As shown in Table 10, our method consistently achieves the best results among all methods on the DAVIS dynamic background subset. The 50-step version reaches the highest SSIM of 0.9828, PSNR of 36.48, and success rate of 91.36, showing clear superiority in both reconstruction and perceptual quality. Traditional methods such as ProPainter and DiffuEraser fall behind in success rate and VQ. Increasing steps improves MiniMax-Remover steadily, confirming the robustness of iterative refinement. Overall, MiniMax-Remover maintains strong temporal consistency while significantly enhancing visual fidelity and task success compared with prior baselines.

Table 9: Comparison of methods on multi-objects videos. TC denotes Temporal Consistency, VQ denotes Visual Quality, and Succ denotes Success Rate.

| Method | SSIM | PSNR | TC | VQ | Succ |
|---|---|---|---|---|---|
| ProPainter [56] | 0.9778 | 35.64 | 0.9773 | 5.67 | 57.41 |
| VideoComposer [48] | 0.8967 | 31.14 | 0.9546 | 2.13 | 9.26 |
| COCOCO [59] | 0.9101 | 32.98 | 0.9523 | 3.37 | 11.11 |
| FLoED [15] | 0.8887 | 32.72 | 0.9691 | 5.09 | 42.59 |
| DiffuEraser [25] | 0.9769 | 35.08 | 0.9790 | 5.81 | 57.41 |
| VideoPainter [2] | 0.9652 | 35.17 | 0.9634 | 5.00 | 24.44 |
| VACE [21] | 0.8949 | 32.67 | 0.9732 | 3.20 | 5.56 |
| MiniMax-Remover (6 steps) | 0.9842 | 36.62 | 0.9773 | 6.41 | 85.18 |
| MiniMax-Remover (12 steps) | 0.9843 | 36.64 | 0.9778 | 6.31 | 87.04 |
| MiniMax-Remover (50 steps) | **0.9847** | **36.69** | **0.9780** | **6.70** | **92.59** |

Table 10: Comparison of different methods on dynamic background videos. TC denotes Temporal Consistency, VQ denotes Visual Quality, and Succ denotes Success Rate.

| Method | SSIM | PSNR | TC | VQ | Succ |
|---|---|---|---|---|---|
| ProPainter [56] | 0.9781 | 35.78 | 0.9778 | 5.53 | 53.09 |
| COCOCO [59] | 0.9125 | 33.26 | 0.9546 | 3.29 | 12.35 |
| FLoED | 0.8892 | 32.75 | 0.9663 | 5.07 | 43.21 |
| DiffuEraser [25] | 0.9741 | 35.12 | 0.9778 | 5.68 | 55.56 |
| VideoComposer [48] | 0.8936 | 31.31 | 0.9531 | 2.13 | 8.64 |
| VideoPainter [2] | 0.9652 | 35.49 | 0.9623 | 3.88 | 19.75 |
| VACE [21] | 0.8942 | 32.72 | 0.9762 | 2.97 | 7.41 |
| MiniMax-Remover (6 steps) | 0.9821 | 36.42 | 0.9786 | 6.16 | 81.48 |
| MiniMax-Remover (12 steps) | 0.9823 | 36.46 | 0.9787 | 6.23 | 86.42 |
| MiniMax-Remover (50 steps) | **0.9828** | **36.48** | **0.9789** | **6.46** | **91.36** |

Table 11: Effect of dilation size on MiniMax-Remover. TC denotes Temporal Consistency, VQ denotes Visual Quality, and Succ denotes Success Rate.

| Dilation | SSIM | PSNR | TC | VQ | Succ |
|---|---|---|---|---|---|
| 2 | 0.9830 | 36.49 | 0.9764 | 6.37 | 91.11 |
| 6 | **0.9847** | **36.66** | 0.9776 | **6.48** | 91.11 |
| 12 | 0.9838 | 36.38 | 0.9778 | 6.40 | **92.22** |
| 24 | 0.9829 | 36.36 | **0.9789** | 6.33 | 84.44 |

## 11.5 Tightness/looseness of the mask

As shown in the Table 11, moderate dilation values yield the best balance between structure and temporal smoothness. Dilation 6 achieves the highest SSIM of 0.9847 and PSNR of 36.66 with strong visual quality, while dilation 12 gives the best success rate of 92.22. Very small or large dilation slightly reduce performance, indicating that moderate spatial context improves reconstruction and temporal consistency.

## 11.6 The Implementation of Learnable Contrastive Tokens

```
1  import torch
2  import torch.nn as nn
3
4  class TokenEmbedding(nn.Module):
5      def __init__(self, dim):
6          super().__init__()
7          self.dim = dim
8          # The first token represents the C+
9          # The second token represents the C-.
10         self.token_embedding = nn.Embedding(2, dim)
11         # using 6 tokens to enhance contrastive tokens.
12         self.mlp = nn.Linear(dim, 6 * dim)
13         self.act_fn = nn.SiLU()
14
15     def forward(self, input_ids):
16         bsz = input_ids.size(0)
17         x = self.act_fn(self.token_embedding(input_ids))
18         x = self.mlp(x).view(bsz, -1, 6, self.dim)
19         return x
20
21 class Stage1_Model(nn.Module):
22     def __init__(self, config):
23         super().__init__()
24         ...
25         self.token_embed = TokenEmbedding(dim)
26         ...
27
28     def forward(self, hidden_states, input_ids):
29         ...
30         tokens = self.token_embed(input_ids)
31         l = hidden_states.shape[1]
32         hidden_states = torch.cat([hidden_states, tokens], dim=1)
33         ...
34         hidden_states = self.attn(query, key, value)[:,:l]
35         ...
36         return hidden_states
```

Pseudo Codes for Stage-1 Model.

## 11.7 The Impact of Learnable Token Number

Table 12: Comparison of different token configurations in stage-1. TC denotes Temporal Consistency, VQ denotes Visual Quality, and Succ denotes Success Rate.

| Method | Tokens | SSIM | PSNR | TC | VQ | Succ |
|--------|--------|------|------|-----|-----|------|
| stage-1 | 1 | 0.9778 | 35.82 | 0.9774 | 6.13 | 66.67 |
| stage-1 | 2 | **0.9801** | **35.88** | **0.9776** | 6.23 | 70.00 |
| stage-1 | 6 | 0.9798 | 35.87 | 0.9773 | 6.39 | **71.11** |
| stage-1 | 8 | 0.9792 | 35.85 | 0.9772 | **6.46** | **71.11** |

As shown in Table 12, using six tokens achieves the best trade-off across all metrics. While the two-token setting slightly improves SSIM and PSNR, the six-token model attains the highest VQ score (6.39) and the best success rate (71.11 %), indicating superior overall quality and stability. Increasing to eight tokens yields no further gain and even reduces SSIM and PSNR, suggesting over-fragmentation. Therefore, the six-token configuration provides the optimal balance between reconstruction fidelity and perceptual consistency.

## 12   Broader Impacts

Minimax-Remover is a diffusion-based model for removing objects from videos in only 6 inference steps without CFG. It enables efficient video editing, film post-production by simplifying complex visual manipulation. However, like all generative models, it could be misused to create deceptive or fake videos, potentially causing misinformation or reputational harm. To mitigate such risks, we will release the model under a CC BY-NC 4.0 license, restricting commercial use and large-scale deployment.

## 13   Limitation and Future Work

In this study, we present a fast and effective method for video object removal that does not rely on CFG. However, our approach has some limitations. First, the object remover in Stage 2 was trained using only 10,000 videos. Second, while the DiT model is efficient, a large portion of the computational overhead comes from the VAE encoding and decoding processes. In future work, we plan to expand the dataset to enhance the remover's robustness and explore the use of a smaller VAE to speed up inference.

