# OpenReview forum: "MiniMax-Remover: Taming Bad Noise Helps Video Object Removal"
_NeurIPS.cc/2025/Conference — NeurIPS 2025 poster_

### Official Review · Reviewer_Hp3M · 2025-06-30

**Clarity:** 3
**Significance:** 3
**Originality:** 3
**Rating:** 5
**Confidence:** 3

**Summary:**

This paper proposes a two stage trained video object removal pipeline on DiT backbone. In the first stage, they point out the ambiguity of the text conditioning for object removal and remove the cross attention layers and then train the model using contrastive tokens ( $c^{+}$ for addition, $c^-$ for regeneration) in self attention removing the need for cfg. In the second stage, they further improve the model by distillation to remove time steps and minimax optimization to improve robustness to adversarial input noise which cause failures.

**Questions:**

- **Contrastive-token validation:** please show evidence showcasing that the tokens really learn distinct “remove” and “regenerate” semantics. Clear proof here would substantially strengthen the paper. Possible evidence can be attention maps, or a quantitative breakdown of removal and hallucination errors without presence of one of the tokens during inference in stage 1.

- **Contrastive-token count:** Author's state using 6 tokens for contrastive c⁺ and c⁻. How does the author's come up with number 6. I assume it is empirical, but no analysis on token count's effect on model performance after stage 1 is provided.

- **Shadow's** In the supplementary material videos, it is stated that the model's does not remove shadows as they are not included in the mask. Does the model successfully remove both the object and the shadow if they are both included in the mask?

- **Bad-noise search analysis:** Does further optimizing the bad adversarial noise for second stage training improve robustness or does it degrade performance? Also how does the author's come up with the finding that failure cases correlation with specific input noise pattens?

- **Broken Link:** In line 783 of the paper, one reference link is broken which results in a **?** in latex. That can be fixed.

**Ethical Concerns:**

["NO or VERY MINOR ethics concerns only"]

**Final Justification:**

I think this paper solves an open problem in literature with a novel finding. In rebuttal, I have asked the authors to justify Contrastive-token validation and bad noise search analysis and they have addressed my concerns. As a result, I have increased my score.

**Limitations:**

- The checklist marks Broader Impacts as provided in appendix , yet I could not find a dedicated section in the appendix. Can the authors please point to the exact paragraph or expand the discussion to cover potential harms (e.g., erasing evidence, facilitating misinformation).

- More failure cases could be discussed. Currently the article only states the dataset and the gpu consumption of the VAE as a limitation. However,  limitations of the model in terms of failure cases can be discussed further with provided examples.

**Quality:**

3

**Strengths And Weaknesses:**

### **Strengths**:
- The author's are addressing a valid problem being present in the current literature. Optical flow warping based model's are fast in inference time but most of the time fail with blurry hallucinated artifacts while the Diffusion based method's are prone to object addition and slow. The proposed method is trying to achieve both fast and high quality object removal.
- The proposed method shows improvements over current literature in terms of metric's such as SSIM, PSNR and TC and authors have conducted a strong literature search for both related work and works to compare.
- Overall the paper is clear and easy to follow. Motivations for each step are explained and diagrams help with understanding the architecture and training.
- Second stage adversarial noise based minimax training is an original finding and contrastive tokens for training “remove vs regenerate” inside self-attention is new to object-removal literature.

### **Weaknesses**:

1. **Overall the claims made in the paper should be further and better supported with examples:**

- Contrastive token's semantics are not directly demonstrated. The paper says c⁺ encourages removal and c⁻ discourages regeneration. In table 3 it shows improvements but there is no clear evidence showing whether c⁺ really encourages filling and c⁻ discourages regeneration or not. It might be the case that these tokens are acting in a different behavior and that is not really proved in the paper.

- Correlation of failure cases after stage 1 with specific input noise pattens, which is the motivation for stage 2 training, needs further in depth explanation. How author's have come up with that finding is not explained.

2.  **Limited visual examples**:
- Only success cases are shown. Failure cases are described verbally but not illustrated with visual examples.
- More visual examples on comparisons with other baselines are needed.

3.  **Statistical robustness missing.**
- All metrics are single-seed; no standard deviations or confidence intervals are reported.

---

> ### Author Rebuttal · Authors · 2025-07-31
>
> Thank you for your valuable comments. Your suggestions are really appreciated and help us further improve the quality of our submission.
>
> Our method is robust and efficient, and it does not rely on external models to precompute complex conditions. Currently, our Minimax-Remover has been open-sourced and has received a lot of attention from the community. We have received many feedbacks, and some users have even contributed by developing ComfyUI code and integrating our method into a widely used ComfyUI node.
>
> ---
>
> ***Since NeurIPS 2025 prohibits external links and updated pdfs, we were unable to provide visualization results during both the rebuttal and discussion stages. We hope for your understanding.***
>
> ---
>
> **Q1: Contrastive token's semantics are not directly demonstrated. The paper says c⁺ encourages removal and c⁻ discourages regeneration. In table 3 it shows improvements but there is no clear evidence showing whether c⁺ really encourages filling and c⁻ discourages regeneration or not. It might be the case that these tokens are acting in a different behavior and that is not really proved in the paper.**
>
> Thank you for your suggestion. Below, we present the comparison results with and without the usage of positive and negative control tokens.
>
> | Method| SSIM |PSNR |TC| VQ| Succ|
> |-----|-----|-----|-----|-----|-----|
> | stage-1-only-postive-tokens   w/o CFG |0.9764|35.67|0.9769|6.36 |58.89|
> | stage-1-only-negative-tokens  w/o CFG |0.9721|34.66|0.9681|6.10 |15.56|
> | stage-1-both-tokens w/ CFG |**0.9798**|**35.87**|**0.9773**|**6.39**|**71.11**|
>
> The above table shows that using both tokens will have better removal results, while only using positive token is not enough.
>
> &nbsp;
>
>
> **Q2: Correlation of failure cases after stage 1 with specific input noise pattens, which is the motivation for stage 2 training, needs further in depth explanation. How author's have come up with that finding is not explained.**
>
> R2: Thank you for your valuable suggestion. We conducted experiments showing that the random seed significantly affects the success of the stage-1 model. We will include these results in the revised paper. By using three different random seeds to inference our stage-1 model three times, we found that only 65.56% has consistent successful results (all three seeds resulted in successful outcomes).
>
> | Method| Succ|
> |---|---|
> | stage-1-seed-0 |71.11|
> | stage-1 seed-1 |72.22|
> | stage-1 seed-2 |75.56|
> |stage-1 success on all three seeds|65.56|
>
> | Method| SSIM |PSNR |TC| VQ| Succ|
> |---|---|---|---|---|---|
> | stage-1|0.9798|35.87|0.9773|6.39 |71.11|
> | stage-2|**0.9847**|**36.66** |**0.9776**|**6.48** |**91.11**|
>
> As shown in the table, the stage-1 model is sensitive to random seed, after fine-tuning with the minimax optimization, the performance has significant increase. These findings support our statement: *"these failure cases are strongly correlated with specific input noise patterns."*
>
> &nbsp;
>
> **Q3：Limited visual examples:**
>
> **Q3.1: Only success cases are shown. Failure cases are described verbally but not illustrated with visual examples.**
>
> R3.1: Thanks for your advice. Since the file size limitation of the openreview, we couldn't put  too many cases in main body. We promise that we will put more visual examples in the appendix of our revised paper.
>
> **Q3.2: More visual examples on comparisons with other baselines are needed.**
>
> R3.2: We will put more visual examples with different results of removal models in our revised paper. We promise to put them in the appendix.
>
> &nbsp;
>
> **Q4: Statistical robustness missing.  All metrics are single-seed; no standard deviations or confidence intervals are reported.**
>
> R4: Thanks for your advice. According to your suggestion, we supplied more experiments with different seeds to show the robustness of our minimax-remover. The results are shown in the below table:
>
> | Method| SSIM|PSNR|TC|VQ|Succ|
> |-----|-----|-----|-----|-----|-----|
> | stage-2 seed-0|**0.9847**|36.66 |0.9776|6.48 |91.11|
> | stage-2 seed-1|0.9846|36.62 |0.9772|**6.54** |91.11|
> | stage-2 seed-2|**0.9847**|**36.69** |**0.9788**|6.53 |**92.22**|
>
> | Method| SSIM|PSNR|TC|VQ|Succ|
> |-----|-----|-----|-----|-----|-----|
> |Mean|0.9847|36.66 |0.9779|6.52 |91.48|
> |Std|4.7e-5|0.028 |0.0680|0.026|0.523|
>
> As shown in the table, the standard variance of success rate is small. It demonstrates that minimax-remover is robust to different random seed.
>
> &nbsp;
>
> **Q5：Contrastive-token validation: please show evidence showcasing that the tokens really learn distinct “remove” and “regenerate” semantics. Clear proof here would strengthen the paper. Possible evidence can be attention maps, or a quantitative breakdown of removal and hallucination errors without presence of one of the tokens during inference in stage 1.**
>
> R5: Thanks for your advice. We highly agree that we should put more experiments in our paper to show the effect of the contrastive tokens. The experiments are shown below:
>
> | Method| SSIM |PSNR |TC| VQ| Succ|
> |-----|-----|-----|-----|-----|----|
> | stage-1-only-postive-tokens w/o CFG |0.9764|35.67|0.9769|6.36 |58.89|
> | stage-1-only-negative-tokens w/o CFG |0.9721|34.66|0.9681|6.10 |15.56|
> | stage-1-both-tokens w/ CFG |**0.9798**|**35.87**|**0.9773**|**6.39** |**71.11**|
>
> The results in the table indicate that the combination of positive and negative control tokens yields improved removal performance.
>
> *Since NeurIPS prohibits external links and updated pdfs during the rebuttal and discussion stages, visualization results such as attention maps will be included in the appendix of our next manuscript version.*
>
> &nbsp;
>
> **Q6: Contrastive-token count**
>
> **Q6.1:  Author's state using 6 tokens for contrastive c⁺ and c⁻. How does the author's come up with number 6.**
>
> R6.1: Thank you for your suggestion.
>
> * Our intention in using 6 control tokens instead of 1 token is to enhance the model’s controllability. In the attention mechanism, using more control tokens increases the overall control weight, as each vision token receives multiple control signals. The experimental results are shown below.
>
> | Method|Tokens| SSIM |PSNR |TC| VQ| Succ|
> |-----|-----|-----|-----|-----|-----|-----|
> | stage-1 |1|0.9778|35.82|0.9774|6.13|66.67|
> | stage-1 |2|**0.9806**|**35.88**|**0.9776**|6.23|70.00|
> | stage-1 |6|0.9798|35.87|0.9773|6.39|**71.11**|
> | stage-1 |8|0.9797|35.85|0.9772|**6.46**|**71.11**|
>
> As shown in the table, using 6 tokens is sufficient to achieve effective control in the removal process. Increasing the number of tokens to 8 does not bring in more improvement.
>
> **Q6.2: no token count's effect on model performance after stage 1 is provided.**
>
> R6.2: We use contrastive tokens only in stage-1. Since stage-2 does not rely on CFG, we remove the contrastive tokens during this stage.
>
> &nbsp;
>
> **Q7: Shadow's in the supplementary material videos. Does the model successfully remove both the object and the shadow if they are both included in the mask?**
>
> R7: Yes, our model is capable of handling such cases. If masks are provided for both the object and its shadow, our model can successfully remove both components. We noticed that community users have provided many successful cases that remove both object and shadow. Below, we present more results of our model on multi-object removal experiments.
>
> | Method| SSIM |PSNR |TC| VQ| Succ|
> |-----|-----|-----|-----|-----|-----|
> |ProPainter |0.9781|35.78|0.9778|5.53|53.09|
> | CoCoCo|0.9125|33.26|0.9546|3.29|12.35|
> |FloED|0.8892|32.75|0.9663|5.07|43.21|
> |DiffuEraser|0.9741|35.12|0.9778|5.68|55.56|
> |VideoComposer|0.8936|31.31|0.9531|2.13|8.64 |
> |VideoPainter|0.9652|35.49|0.9623|3.88|19.75|
> |Vace |0.8942|32.72|0.9762|2.97|7.41 |
> |MiniMax-Remover (6 steps) |0.9821|36.42|0.9786|6.16|81.48|
> |MiniMax-Remover (12 steps)|0.9823|36.46|0.9787|6.23|86.42|
> |MiniMax-Remover (50 steps)|**0.9828**|**36.48**|**0.9789**|**6.46**|**91.36**|
>
> &nbsp;
>
> **Q8: Bad-noise search analysis**
>
> **Q8.1: Does further optimizing the bad adversarial noise for stage-2 training improve robustness or degrade performance?**
>
> R8.1: In our earlier experiments, we attempted to use a two-step optimization process to generate adversarial noise. While this approach requires more time to converge, it did not lead to noticeable improvements in visual results.
>
> **Q8.2: How does you find that failure cases correlation with specific input noise pattens?**
>
> R8.2: During stage-1 evaluation, we observed that changing the random seed often flipped failure cases into successes and vice versa. This frequent phenomenon suggests that the outcome of removal is highly sensitive to input noise.*.
>
> This observation inspired us to further investigate the characteristics of **"bad" noise**. To address its adverse effects, we proposed a **minimax optimization strategy** to mitigate the impact of such noise. The results obtained in stage-2 significantly outperform those in stage-1, which further supports our claim. *The experimental results can be referred to our response to Q2.*
>
> &nbsp;
>
> **Q9: Broken Link: In line 783 of the paper, one reference link is broken which results in a ? in latex. That can be fixed.**
>
> R9: Thanks for pointing out this typo. We apologize for the mistake and will correct it in the revised paper.
>
> &nbsp;
>
> **Q10: Disscussing broader impacts in appendix.**
>
> R10: We appreciate your valuable suggestion. A discussion on the broader impacts will be added in the revised version of the paper.
>
> &nbsp;
>
> **Q11: More failure cases could be discussed. The failure cases can be discussed further with provided examples.**
>
> R11: Thanks for your advice. We will put failure examples in the appendix. However, due to the policy of NeurIPS, we couldn't provide links or renewed pdf to show the visualization results. We sincerely hope your understanding.
>
> ---
>
> **We really hope our response has resolved your concerns, and we would greatly appreciate it if you could consider increasing your score.**

---

> > ### Comment · Reviewer_Hp3M · 2025-08-04
> >
> > Thank you for your responses, which have addressed my concerns. I will raise my score to a 5 and hope for acceptance. I would also like to thank the authors for conducting extensive number of experiments in a short amount of time.

---

> ### Author Response · Authors · 2025-08-04
>
> Thank you very much for your kind feedback and for increasing the score! We sincerely appreciate your thoughtful comments and are grateful for your decision to raise the score.

---

### Official Review · Reviewer_oy1i · 2025-06-30

**Clarity:** 3
**Significance:** 3
**Originality:** 3
**Rating:** 5
**Confidence:** 4

**Summary:**

This paper proposes MiniMax-Remover, a two-stage approach for video object removal. In stage 1, the cross-attention layers are removed from the pretrained video generation model Wan2.1-1.3B. Instead, the learnable contrastive condition tokens are added to the module. Stage 2 employs a minimax optimization strategy as the inner maximization searches for adversarial bad noise, while the outer minimization trains the model to be robust against such noise. Experimental results show that the proposed method achieves state-of-the-art results with only 6 sampling steps.

**Questions:**

- How does the method perform on videos with dynamic backgrounds or multiple overlapping objects?
- In the ablation study, the impact of positive or negative tokens should be further explored, such as removing one and evaluating performance degradation.
- In line 139, why split the feature into 6 tokens to increase control ability? How to choose the number of tokens?
- In line 185, how to select 10K video pairs manually from the 17K generated by the Stage 1 model?
- In Table 3, the GPT-O3 is used for evaluation. How to verify that large models could replace manual scoring?

**Ethical Concerns:**

["NO or VERY MINOR ethics concerns only"]

**Final Justification:**

Since the author's response and additional experiments answered all my questions, I recommend accepting this paper. The specific reasons are as follows:
- The authors demonstrate the effectiveness of their method in videos with dynamic backgrounds through additional experiments, acknowledging the challenges existing methods face in handling multiple overlapping objects.
- The authors further refine their ablation experiments.
- The authors demonstrate the practicality of using GPT-O3 for evaluation.

**Limitations:**

yes

**Quality:**

3

**Strengths And Weaknesses:**

Strengths:
- The paper is well organized and well written.
- The idea is interesting and effective. By eliminating CFG and reducing sampling steps, the method significantly speeds up inference while maintaining the quality.
- Experimental results show the superiority of the proposed method.

Weaknesses:
- Figure 3 is not cited in the main text.
- In stage 2, the model is only trained on 10K manually curated videos, which may limit its generalization to real-world scenarios.
- Since stage 2 of the method is designed to address failure cases, more failure case analyses should be made to present the limitations of MiniMax-Remover.

---

> ### Author Rebuttal · Authors · 2025-07-31
>
> Thank you very much for your recognition. As you mentioned, our method is *interesting and effective*. The experiments demonstrate that **minimax-remover** generalizes well on both DAVIS and Pexels datasets, as evaluated by both LLMs and human annotators.
>
> Moreover, our **minimax-remover** has been widely embraced by the community. It has been integrated into a popular ComfyUI node by community users, making it easily accessible even to those without a deep learning background. Interestingly, some users have started applying it to produce high-quality videos. Remarkably, it performs well on 1080P videos and can even handle videos with up to 500 frames.
>
> ---
>
> ***Due to NeurIPS' policy this year prohibiting updates to the pdf and the inclusion of external links, we are unable to put the visualization results in both rebuttal and discussion stage. We sincerely hope for your understanding.***
>
> ---
>
> **Q1: Figure 3 is not cited in the main text.**
>
> R1: Thank you for your reminder. We will include a proper citation for this figure in the main text. We really appreciate your helpful suggestion.
>
> &nbsp;
>
> **Q2: In stage 2, the model is only trained on 10K manually curated videos, which may limit its generalization to real-world scenarios.**
>
> R2: Thanks for pointing this problem.
>
> * **In stage-1, our model has been trained on large scale dataset with 2.5 millions of videos.**  Stage 2 can be viewed as the supervised fine-tuning (SFT) stage, similar to that in LLM training. With the usage of minimax optimization, even a small-scale training dataset can lead to significant performance improvements and better alignment with human preferences.
>
> * **Community users have reported it has good generalization in the real world.** Many community users have tried our open-sourced model and reported strong generalization capabilities across a variety of videos. As shown in the table below, applying minimax optimization with only 10K videos yields substantial improvements compared to stage-1 model.
>
> | Method| SSIM |PSNR |TC| VQ| Succ|
> |--------|-----|-----|-----|-----|-----|
> | stage-1|0.9798|35.87|0.9773|6.39|71.11|
> | stage-2|**0.9847**|**36.66** |**0.9776** |**6.48** |**91.11**|
>
> As shown in the table, the fine-tuned stage-2 model on 10K data significantly improves the successful rate by 20%.
>
> &nbsp;
>
> **Q3: How does the method perform on videos with dynamic backgrounds or multiple overlapping objects?**
>
> R3： Thanks for your comments!
>
> * For Dynamic background, we find that around 90% videos in DAVIS dataset are *not captured with fixed camera position*. Therefore, we selected videos with the dynamic background and evaluate our method and baselines on these videos. The following table shows the evaluation results of our methods and the previous methods.
>
>
> | Method| SSIM |PSNR |TC| VQ| Succ|
> |--------|-----|-----|-----|-----|-----|
> |	ProPainter                |0.9781|35.78|0.9778|5.53|53.09|
> | CoCoCo                    |0.9125|33.26|0.9546|3.29|12.35|
> |	FloED                     |0.8892|32.75|0.9663|5.07|43.21|
> |	DiffuEraser               |0.9741|35.12|0.9778|5.68|55.56|
> | VideoComposer             |0.8936|31.31|0.9531|2.13|8.64 |
> | VideoPainter              |0.9652|35.49|0.9623|3.88|19.75|
> | Vace                      |0.8942|32.72|0.9762|2.97|7.41 |
> | MiniMax-Remover (6 steps) |0.9821|36.42|0.9786|6.16|81.48|
> | MiniMax-Remover (12 steps)|0.9823|36.46|0.9787|6.23|86.42|
> | MiniMax-Remover (50 steps)|**0.9828**|**36.48**|**0.9789**|**6.46**|**91.36**|
>
> Results on the above table show that minimax-remover is robust to dynamic background.
>
> * The removal of multiple overlapping objects remains a challenging problem for video object removal systems. It is influenced by several factors, such as the generative capacity of the base model and the complexity of high-quality videos with densely overlapping content. Our minimax-remover demonstrates promising performance on this task, although some failure cases still occur.
>
> &nbsp;
>
> **Q4: In the ablation study, the impact of positive or negative tokens should be further explored, such as removing one and evaluating performance degradation.**
>
> R4: Thanks for your advice. Here we evaluate the impact of positive or negative tokens. The results are shown below:
>
> | Method| SSIM |PSNR |TC| VQ| Succ|
> |--------|-----|-----|-----|-----|-----|
> | stage-1-only-postive-tokens   w/o CFG |0.9764|35.67|0.9769|6.36 |58.89|
> | stage-1-only-negative-tokens  w/o CFG |0.9721|34.66|0.9681|6.10 |15.56|
> | stage-1-both-tokens w/ CFG            |**0.9798**|**35.87**|**0.9773**|**6.39**|**71.11**|
>
> As indicated in the table, utilizing only positive tokens results in fair performance, whereas relying solely on negative tokens leads to a noticeable performance decline. The incorporation of CFG further improves the model’s performance. These findings highlight the importance of both positive and negative tokens in the stage-1 model.
>
> &nbsp;
>
> **Q5: In line 139, why split the feature into 6 tokens to increase control ability? How to choose the number of tokens?**
>
> The motivation for using 6 tokens is to enhance the model's controllability. In the attention computation mechanism, employing more control tokens results in greater control ability, as each vision token receives 6 separate control signals. The following presents the evaluation results for different numbers of control tokens.
>
> | Method|Tokens| SSIM |PSNR |TC| VQ| Succ|
> |--------|-----|-----|-----|-----|-----|-----|
> | stage-1 |1         |0.9778|35.82|0.9774|6.13|66.67|
> | stage-1 |2         |**0.9801**|**35.88**|**0.9776**|6.23|70.00|
> | stage-1        |6  |0.9798|35.87|0.9773|6.39|**71.11**|
> | stage-1 |        8 |0.9792|35.85|0.9772|**6.46**|**71.11**|
>
> Overall, using 6 tokens yields a higher success rate compared to using only 1 token, while increasing the number of tokens beyond 6 does not lead to a significant performance improvement.
>
> &nbsp;
>
> **Q6: In line 185, how to select 10K video pairs manually from the 17K generated by the Stage 1 model?**
>
> R6：We asked three human annotators to filter the videos using a voting mechanism. Afterwards, each annotator independently reviewed the videos again to confirm that there were no artifacts, blurriness, or unwanted objects in the masked regions. Together, these two steps ensure the overall quality of the data.
>
>
> &nbsp;
>
> **Q7: In Table 3, the GPT-O3 is used for evaluation. How to verify that large models could replace manual scoring?**
>
> R7: Thank you for your comment. Here, we present the evaluation results for the LLMs. We asked human annotators to select 100 videos, consisting of 50 failure cases and 50 success cases. The evaluation results are shown below.
>
> * Evaluation for different LLMs.
>
> | Success Rate         | GPT-4O| GPT-4O-mini| GPT-4-turbo | GPT-O3    |
> |----------------------|-------|------------|-------------|-----------|
> | Acc in Failure Cases | 94.0% | 98.0%      | 86.0%       | **96.0%** |
> | Acc in Success Cases | 88.0% | 26.0%      | 58.0%       | **94.0%** |
> | Overall Accuracy     | 91.0% | 62.0%      | 72.0%       | **95.0%** |
>
> As shown in the table, GPT-O3 achieves the best performance, with a 95% consistency with human annotation. This high level of alignment suggests that the decisions made by GPT-O3 can be considered reasonable.
>
> ---
>
>
> We sincerely thank you for your efforts in reviewing our paper. Your valuable comments helps us improve the quality of the manuscript.
>
> **We hope our responses have addressed your concerns and you will reconsider your score accordingly.**

---

> > ### Comment · Reviewer_oy1i · 2025-08-01
> >
> > Thank you for your reply and additional experiments, which have answered all my concerns. I will increase the score to accepted.

---

> ### Author Response · Authors · 2025-08-02
>
> Thank you very much for your valuable comments and for raising your evaluation of our manuscript! We are pleased to see our work recognized and really appreciate your thoughtful review.

---

### Official Review · Reviewer_fNcQ · 2025-07-02

**Clarity:** 3
**Significance:** 3
**Originality:** 3
**Rating:** 5
**Confidence:** 3

**Summary:**

This paper proposes "MiniMax-Remover," a novel method designed to enhance the performance and efficiency of video object removal. The authors begin by highlighting three key problems associated with existing video object removal methods:

- Generation of undesired content or artifacts, including occlusions and blurs within masked regions.
- Heavy reliance on auxiliary priors, such as optical flow, text prompts, or DDIM inversion.
- Requirement of numerous sampling steps for achieving high visual fidelity, coupled with dependence on CFG, thus demanding significant computational resources.

To overcome these limitations, the authors propose "MiniMax-Remover," utilizing the pretrained Wan2.1-1.3B model as the foundation. They eliminate textual conditioning and all cross-attention layers from the pretrained DiT Block, effectively removing the dependency on auxiliary priors. The MiniMax-Remover method is structured around a two-stage training framework:

- Stage 1 employs contrastive conditioning tokens (c+, c-) injected into self-attention layers to explicitly teach the model the distinction between object removal and object generation, enhancing model clarity and effectiveness.

- Stage 2 involves further refining the model through a min-max optimization strategy:
  - Stage 2-1 focuses on identifying and selecting challenging or "bad noise" scenarios that lead to model errors.
  - Stage 2-2 utilizes a dataset of 17K videos from stage 1 results, consisting of 10K successfully removed object videos and 7K original videos, to perform adversarial training. The model is thus trained robustly to handle challenging inputs.

Overall, this approach significantly enhances video object removal performance, delivering high-quality results with improved computational efficiency.

**Questions:**

1. In Section 3.3, "we observe that these failure cases are strongly correlated with specific input noise patterns”, could you provide experimental evidence or additional analysis supporting this observation? Clarifying this point could significantly improve the understanding and validation of your proposed method.
2. The explanation regarding the learnable contrastive tokens is somewhat unclear. Could you elaborate on the token embedding layer used and clarify how exactly these tokens are learned during training?
3. In the supplemental video materials, objects appear to be effectively removed, but shadows remain visible. Can the method be extended or improved to naturally remove shadows along with the objects?

**Ethical Concerns:**

["NO or VERY MINOR ethics concerns only"]

**Final Justification:**

I think this paper not only addresses existing issues in video object removal but also proposes the most lightweight model to date. In the rebuttal, I inquired about the experimental analysis of bad noise as well as the limitations of the training data, and I received responses to both. Since my concerns have been addressed, I have decided to change my score from the original rating to Accept.

**Limitations:**

yes

**Paper Formatting Concerns:**

No major formatting issues observed

**Quality:**

3

**Strengths And Weaknesses:**

### Strengths

- The paper is well-structured and clearly presented, making it accessible to readers.
- Effectively leverages a lightweight DiT-based architecture, optimizing for both computational efficiency and practical performance.
- Particularly noteworthy is the method’s ability to perform efficiently without relying on CFG, using only 6 sampling steps. As illustrated in Table 1, the proposed model has the smallest number of parameters, fastest inference, and lowest GPU memory usage, making it highly practical.
- Provides a comprehensive ablation study clearly delineating individual contributions of each training stage (Stage 1 and Stage 2), offering valuable insights into the importance and effectiveness of each component.

### Weaknesses

- In Section 3.3, the claim "we observe that these failure cases are strongly correlated with specific input noise patterns" lacks robust experimental validation, making it challenging to conclusively support this statement.
- Limitation of training data: The Stage 2 training relies heavily on manually selected data, presenting significant limitations:
    - Manually selecting 10,000 successful object removal videos requires substantial human resources, making it expensive and difficult to scale.
    - The definition of "high-quality" is inherently subjective, potentially introducing biases based on individual annotators’ preferences or criteria, thereby influencing the model’s overall performance.

---

> ### Author Rebuttal · Authors · 2025-07-31
>
> Thanks for your recognition of our work. As you mentioned, our paper *significantly enhances video object removal performance, delivering high-quality results with improved computational efficiency*. Your valuable comments help us to further improve our paper. The following are our responses to your questions.
>
> ---
>
> ***Please note that, in accordance with this year's NeurIPS guidelines, external links and updated PDFs are not permitted during the rebuttal and discussion stages. Consequently, we are unable to share an updated pdf or visualization results.***
>
> ---
>
> **Q1: "We observe that these failure cases are strongly correlated with specific input noise patterns" lacks robust experimental validation, making it challenging to conclusively support this statement.**
>
> R1: Thanks for your comments.
>
> * When evaluating the model trained in Stage-1, we observed that simply changing the random seed often transformed failure cases into successful ones, or verse vice. This phenomenon occurred frequently. Based on this observation, we found that * the success or failure of the removal results is highly correlated with input noise*.
>
> * This observation inspired us to further investigate the characteristics of **"bad" noise**. To address its adverse effects, we proposed a **minimax optimization strategy** to mitigate the impact of such noise. The results obtained in Stage-2 significantly outperform those in Stage-1, which further supports our claim.
>
> * After receiving your comments, we conducted additional experiments to verify our claim. Specifically, we evaluated the Stage-1 model using three different random seeds (i.e., 0, 1, and 2).
>
> We found that only 65.56% has consistent successful results (all three seeds resulted in successful outcomes).
>
>
> | Method| Succ|
> |--------|-----|
> | stage-1-seed-0       |71.11|
> | stage-1 seed-1       |72.22|
> | stage-1 seed-2       |75.56|
> |stage-1 success on all three seeds|65.56|
>
> As shown in the above table, the stage-1 model is sensitive to random seed.
>
> &nbsp;
>
> **Q2: Manually selecting 10,000 successful object removal videos requires substantial human resources, making it expensive and difficult to scale**
>
> R2:  Thank you for your insightful comment. We would like to clarify that this method can scale up to large-scale datasets by the following two steps. First, we can apply our Stage-2 model to a large amount of videos and leverage GPT-O3 to identify and exclude failure cases. Second, we train a new remover model on the filtered data.
>
> &nbsp;
>
> **Q3: The definition of "high-quality" is inherently subjective, potentially introducing biases based on individual annotators’ preferences or criteria, thereby influencing the model’s overall performance.**
>
> R3: We asked three human annotators to filter the videos multiple times using a voting mechanism. Subsequently, each annotator reviewed the videos again to ensure that there were no artifacts, blurriness, or undesired objects in the masked regions. These two steps collectively ensure the quality of the data.
>
> &nbsp;
>
> **Q4: In Section 3.3, "we observe that these failure cases are strongly correlated with specific input noise patterns”, could you provide experimental evidence or additional analysis supporting this observation? Clarifying this point could significantly improve the understanding and validation of your proposed method.**
>
> R4: Thank you for your valuable suggestion. We conducted experiments showing that the random seed significantly affects the success of the stage-1 model. We will include these results in the revised paper. Using three random seeds for object removal, we found that only 65.56% has consistent successful results (all three seeds resulted in successful outcomes).
>
>
> | Method| Succ|
> |--------|-----|
> | stage-1-seed-0       |71.11|
> | stage-1 seed-1       |72.22|
> | stage-1 seed-2       |75.56|
> |stage-1 success on all three seeds|65.56|
>
> | Method| SSIM |PSNR |TC| VQ| Succ|
> |--------|-----|-----|-----|-----|-----|
> | stage-1|0.9798|35.87|0.9773|6.39 |71.11|
> | stage-2|**0.9847**|**36.66** |**0.9776**|**6.48** |**91.11**|
>
> As shown in the table, the stage-1 model is sensitive to random seed, after fine-tuning with the minimax optimization that aims to tame the "bad noise", the performance has significant increase. These two findings support our statement: *"these failure cases are strongly correlated with specific input noise patterns."*
>
>
> &nbsp;
>
> **Q5：The explanation regarding the learnable contrastive tokens is somewhat unclear. Could you elaborate on the token embedding layer used and clarify how exactly these tokens are learned during training?**
>
> R5：Thanks for your reminder. Here, we provide the detailed analysis to our learnable contrastive tokens.
>
> * We implement the `TokenEmbedding` class using `nn.Embedding`, a `SiLU` activation function, and a `Linear` layer. This class is then instantiated within the `Stage1_Model` class. During training, the `hidden_states` are concatenated with contrastive tokens and processed through a self-attention mechanism. Backpropagation is applied to compute gradients for each layer, and the AdamW optimizer is used to update the weights, with a learning rate of `1e-5`.
>
>
> * The simplified code is shown below:
>
> ```python
> import torch
> import torch.nn as nn
>
> class TokenEmbedding(nn.Module):
>     def __init__(self, dim):
>         super().__init__()
>         self.dim = dim
>         # The first dimension represents the c^+
>         # The second dimension represents the c^-.
>         self.token_embedding = nn.Embedding(2, dim)
>         # using more tokens to expand the influence of the contrastive tokens.
>         self.mlp = nn.Linear(dim, 6 * dim)
>         self.act_fn = nn.SiLU()
>
>     def forward(self, input_ids):
>         bsz = input_ids.size(0)
>         x = self.act_fn(self.token_embedding(input_ids))
>         x = self.mlp(x).view(bsz, -1, 6, self.dim)
>         return x
> ```
>
> ```python
> class Stage1_Model(nn.Module):
>     def __init__(self, config):
>         super().__init__()
>         ...
>         self.token_embed = TokenEmbedding(dim)
>         ...
>
>     def forward(self, hidden_states, input_ids):
>         ...
>         tokens = self.token_embed(input_ids)
>         l = hidden_states.shape[1]
>         hidden_states = torch.cat([hidden_states, tokens], dim=1)
>         ...
>         hidden_states = self.attn(query, key, value)[:,:l]
>         ...
>         return hidden_states
> ```
>
> &nbsp;
>
> **Q6: In the supplemental video materials, objects appear to be effectively removed, but shadows remain visible. Can the method be extended or improved to naturally remove shadows along with the objects?**
>
> R6: The answer is yes.
>
> * Our model can remove the shadow if the mask of shadow is given. We want to clarify that our model has the multi-objects removal ability. Many contributors in the community have already made many cases on the Internet.
>
> * Honestly speaking, current model can remove shadow only if the mask of shadow is given. *We believe this is a data-intensive problem, if well-made video pairs are provided, our minimax-remover can be extended to naturally remove shadows along with the objects.*
>
>
> ---
>
> We sincerely appreciate your thoughtful feedback and the time you dedicated to reviewing our work. We hope that our responses and the clarifications have addressed your concerns effectively.
>
> **We kindly ask you to reconsider your score based on our extensive experiments  and clarification. Your reconsideration means a lot to us.**

---

> > ### Comment · Reviewer_fNcQ · 2025-08-08
> >
> > Thank you for your responses and for providing detailed answers, especially to Q1, Q4, and Q5.
> >
> > My concerns have been addressed. In particular, I look forward to seeing the points related to Q5 reflected in the final manuscript. I will change my score to Accept.

---

> ### Author Response · Authors · 2025-08-08
>
> Dear Reviewer fNcQ,
>
> Thank you for the valuable time and the effort in reviewing our paper.
>
> We hope you might find our responses satisfactory. We sincerely hope you will reconsider your rating based on our clarification in responses. Thank you for your time!
>
> Sincerely,
>
> Authors of submission 4051

---

> ### Author Response · Authors · 2025-08-08
>
> Thank you very much for your thoughtful feedback and for changing the score to Accept! We truly appreciate your recognition of our responses, particularly to Q1, Q4, and Q5. In the revised manuscript, we will ensure that the points related to Q5 are fully and clearly incorporated, along with further refinements to address all your comments comprehensively.

---

### Official Review · Reviewer_KYyp · 2025-07-03

**Clarity:** 3
**Significance:** 2
**Originality:** 2
**Rating:** 4
**Confidence:** 4

**Summary:**

**MiniMax-Remover**

The authors propose a model for video object removal which addresses a major limitation of existing methods: slow inference time. They claimthis is mainly due to expensive sampling and the use of classifier free guidance.

The method includes two stages, where the authors train an object removal model in stage 1 by removing the CA layers and introducing contrastive condition tokens to replace textual guidance. Since this still uses CFG with 50 timesteps and generates artifacts, the authors enhance the model in stage 2 improve editing quality and inference speed.

Overall, the paper is well written and a good read.

**Questions:**

- can this method do multi object removal? what if the binary mask covers multiple objects?
- will the code and the trained model be made public?

**Ethical Concerns:**

["NO or VERY MINOR ethics concerns only"]

**Final Justification:**

The additional experiments have answered my questions. Thus, I'm increasing my score. Hope the authors add these additional results to the updated paper.

**Limitations:**

please refer to weaknesses

**Quality:**

3

**Strengths And Weaknesses:**

*Strengths*
- No text prompts required to the model.
- Lightweight as the cross-attention layers are removed, making it efficient
- Requires very few sampling steps (~6) to generate high quality results compared to ~50 in general.

*Weaknesses*
- Although the model doesn't depend on thre text prompt, the mask in L231 is generated using text prompts, so the method indirectly depends on the text prompt.
- Why have no video metrics been used except the temporal consistency metric? Additionally, bakcground removal metrics such as ReMOVE (CVPRW24; A Reference-free Metric for Object Erasure) can be used to evaluate inpainting efficacy per-frame.
- Can the "success-rate" from a model like GPT-03 be trusted? especially in cases when there are multiple similar objects in the scene?
- Looks like all the methods have very high TC and even a model with very low success rate also has a very high temporal consistency, which is weird.
- From the video in Figure 1, we can observe that the model does in remove shadows (branches on the top-right video), reflections (Figure 4) or add in additional information (lighting in the bottom left video). Some of the added content also seems slightly blurry, and could impact quality at higher resolutions.
- no mention about multi-object removal.
- the failures of this method have not been discussed. are there no cases where random objects have been added post stage 2?
- There is no ablation done on the tightness/looseness of the mask (as SAM generates very tight masks) to simulate real-world scenarios.

---

> ### Author Rebuttal · Authors · 2025-07-31
>
> We sincerely thank you for your time and effort in reviewing our paper. Your comments are valuable in helping us improve the quality of our work.
>
> * *The Minimax-Remover adopts a two-stage training strategy. In Stage 1, we train a basic remover using contrastive conditions. In Stage 2, we apply minimax optimization to effectively tame the bad noise. This approach leads to a more stable and faster video object remover.*
>
> * *We have open-sourced both the code and model weights, which are now available on mainstream opensource platforms. Since its release, the Minimax-Remover has received considerable attention from the community. Notably, users have developed and integrated it into a well-known ComfyUI node.*
>
> ---
>
> ***Kindly note that, in accordance with NeurIPS guidelines, we are not permitted to submit external links or updated pdfs of the paper during the rebuttal and discussion stages. Therefore, we are only able to provide experimental results at this time.***
>
> ---
>
> **Q1: The method indirectly depends on the text prompt.**
>
> R1: We would like to clarify that the text prompt is only used during the data preparation phase to construct the training dataset with video-mask pairs. The prompt is not involved in either the training process or the inference stage.
>
>
> &nbsp;
>
> **Q2: Why have no video metrics been used except the temporal consistency metric? Bakcground removal metrics (ReMOVE, CVPRW24) can be used to evaluate inpainting efficacy per-frame.**
>
> R2: Thank you for your valuable suggestion!
>
> * Besides the temporal consistency, we use the SSIM and PSNR to measure the background preservation, and GPT-O3 to evaluate the frames in video.
>
> * We appreciate your insightful suggestion to include ReMOVE evaluation metric, which we had not previously incorporated. After receiving your feedback, we evaluated our Minimax-Remover using this metric. The results are shown below:
>
> | Method           | ReMOVE (w/o Crop) | ReMOVE (w/ Crop) |
> |------|------|------|
> | Vace             | 0.5731            | 0.7723           |
> | ProPainter       | 0.8634            | 0.9259        |
> | FloED            | 0.6950            | 0.8103          |
> | DiffuEraser      | 0.8621            | 0.9212        |
> | CoCoCo           | 0.8009            | 0.8994       |
> | VideoComposer    | 0.6605            | 0.7429  |
> | VideoPainter     | 0.6955            | 0.8279      |
> | ours-6 steps     | 0.8831            | 0.9401       |
> | ours-12 steps    | 0.8843            | 0.9405      |
> | **ours-50 steps**| **0.8850**        | **0.9416**  |
>
> As shown in the table, we can find that our method outperforms all baselines according to the ReMOVE metric.
>
> &nbsp;
>
> **Q3： Can the "success-rate" from a model like GPT-O3 be trusted? especially in cases when there are multiple similar objects in the scene?**
>
> R3: Thanks for pointing out this problem.
>
> * We used different LLMs to evaluate the removal results, including GPT-4O, GPT-4O-mini, GPT-4-turbo and GPT-O3, on 100 removal samples with 50 success samples and 50 failure samples annotated by human. We use the mask to highlight the removed region to let the LLM know where to focus.  **Notably, we find GPT-O3 has the highest consistency with human annotation.** The results are shown below:
>
> | Success Rate         | GPT-4O| GPT-4O-mini| GPT-4-turbo | GPT-O3    |
> |----------------------|-------|------------|-------------|-----------|
> | Acc in Failure Cases | 94.0% | 98.0%      | 86.0%       | **96.0%** |
> | Acc in Success Cases | 88.0% | 26.0%      | 58.0%       | **94.0%** |
> | Overall Accuracy     | 91.0% | 62.0%      | 72.0%       | **95.0%** |
>
> * We also choose 100 multiple objects that removed from video to test the LLM, with 50 success cases and 50 failure cases.
>
> | Success Rate   | GPT-4O    | GPT-4O-mini | GPT-4-turbo | GPT-O3    |
> |----------------------|-----------|-------------|-------------|-----------|
> | Acc in Failure Cases | 90.0%     | 92.0%       | 86.0%       | **98.0%** |
> | Acc in Success Cases | **90.0%** | 48.0%       | 48.0%       | 86.0%     |
> | Overall Accuracy     | 90.0%     | 70.0%       | 67.0%       | **92.0%** |
>
> The table demonstrates that GPT-O3 demonstrates high consistency with human annotations, rather than other LLMs.
>
>
> &nbsp;
>
> **Q4: High Temporal-Consistency (TC) but very low success rate.**
>
> R4:  We want to clarify that the two metrics are not contradictory.
>
> *	The first metric evaluate the motion smoothness of the video.
>
> *	The second metric evaluation is an overall metric, which evaluates motion smoothness, artifacts, blur and undisired objects. A video with blur or undesired objects can also achieve high Temporal-Consistency.
>
> *For example, a failure removal case that add undesired object can have good temporal consistency.* LLM will mark this video as failure case but it achieves a high TC score.
>
>
> &nbsp;
>
> **Q5: The model does not remove shadows. The video has slight blur.**
>
> R5: Thank you for your comments.
>
> We would like to clarify that:
>
> * The shadow issue arises because we do not provide a shadow mask or reflection mask to our model. This problem can be solved by providing both object and shadow/reflection masks to the removal model.
>
> * Upon careful inspection of the original frames, we observed that the lighting artifact is present even before the removal process and outside the masked region. This suggests that it is not an artifact introduced by removal, but rather a persistent feature of the background.
>
> * As for the slight blur, we observed that it does not impact the visual quality at normal resolution. Performing inference at a higher resolution can produce higher-quality results.
>
>
>
> &nbsp;
>
>
> **Q6: Multi-Object Removal.**
>
> We would like to point out that our method performs effectively on the **DAVIS** dataset, where approximately **60%** of the videos contain multi-object masks. Additionally, the demo video provided in the supplementary materials includes several examples of multi-object removal. We provide the evaluation on multi-objects videos as shown in the following.
>
> *Some Multi-Object Removal Examples of Demo Video in Supplementary File*
>
> - **1:40 left** – a stick and a flower pot
> - **1:40 right** – two people
> - **1:50 right** – the masks related to the person are separated
> - **2:14 bottom** – multiple people boxing
> - **2:34 top** – multiple goldfish
> - **2:34 bottom** – person and backpack
> - **2:40 top** – koala’s hands and body are separate masks
> - **2:40 bottom** – surfboard, person, and rope/line
> - **2:46 bottom** – a crowd of people
> - **2:52 top** – person and motorcycle
> - **2:52 bottom** – person and motorcycle
> - **2:56 top** – person and skateboard
>
> *Multi-Object Evaluation Results*
>
> | Method                     | SSIM   | PSNR  | TC     | VQ   | Succ   |
> |----------------------------|--------|-------|--------|------|--------|
> | ProPainter                 | 0.9778 | 35.64 | 0.9773 | 5.67 | 57.41  |
> | VideoComposer              | 0.8967 | 31.14 | 0.9546 | 2.13 | 9.26   |
> | CoCoCo                     | 0.9101 | 32.98 | 0.9523 | 3.37 | 11.11  |
> | FLoED                      | 0.8887 | 32.72 | 0.9691 | 5.09 | 42.59  |
> | DiffuEraser                | 0.9769 | 35.08 | 0.9790 | 5.81 | 57.41  |
> | VideoPainter               | 0.9652 | 35.17 | 0.9634 | 5.00 | 24.44  |
> | VACE                       | 0.8949 | 32.67 | 0.9732 | 3.20 | 5.56   |
> | MiniMax-Remover (6 steps)  | 0.9842 | 36.62 | 0.9773 | 6.41 | 85.18  |
> | MiniMax-Remover (12 steps) | 0.9843 | 36.64 | 0.9778 | 6.31 | 87.04  |
> | MiniMax-Remover (50 steps) | **0.9847** | **36.69** | **0.9780** | **6.70** | **92.59** |
>
> As shown in the table, MiniMax-Remover consistently outperforms all baselines across all metrics in the multi-object removal setting.
>
>
> &nbsp;
>
> **Q7: The failures of this method have not been discussed. Are there no cases where random objects have been added post stage 2?**
>
> R7: Thanks for your advice. We will put the failure cases in our paper in the revised version. Generally speaking, our minimax-remover is robust in most scenerios. However, we observe it has higher failure rate in extreme long video, such as videos with over than 500 frames. Moreover, it has some failure rate in extreme loose masks (e.g., looseness=32 dilation).
>
> &nbsp;
>
> **Q8: Ablation done on the tightness/looseness of the mask.**
>
> Thanks for your advice, we have supplied the tightness/looseness of the mask in the following table. The higher dilation hyperparameter means looser masks.
>
> | Method| Dilation|SSIM |PSNR |TC| VQ| Succ|
> |--------|-----|-----|-----|-----|-----|-----|
> | MiniMax-Remover| 2   |0.9830|36.49|0.9764|6.37|91.11|
> | MiniMax-Remover| 6   |**0.9847**|**36.66**|0.9776|**6.48**|91.11|
> | MiniMax-Remover| 12  |0.9838|36.38|0.9778|6.40|**92.22**|
> | MiniMax-Remover| 24  |0.9829|36.36|**0.9789**|6.33|84.44|
>
> As shown in the table, our minimax-remover performs well in the reasonable masks. However, when giving it extreme loose masks, it will have higher failure rate.
>
>
>
> &nbsp;
>
> **Q9: Can this method do multi object removal? what if the binary mask covers multiple objects?**
>
> R9: Definately, our method can do multi-object removal. You can find the multiple object removal in our demo video from the supplementary file. Moreover, we also evaluate on the videos with multi-objects in DAVIS dataset. **Please refer to our response to Q6 for our experiments and explanation.**
>
>
> &nbsp;
>
> **Q10： Will the code and the trained model be made public?**
>
> R10：Sure, the code and trained model have been opensourced on github and huggingface. We provide the gradio demo with SAM2, users can click single object or multiple objects and then remove it from the video. Our minimax-remover receives a lot of attention from the public. It has been integrated in the ComfyUI node and is popular in the commuity.
>
> ---
>
> **We sincerely hope you can reconsider your ratings based on our response. Your comments can highly help us to improve our submitted paper.**

---

> > ### Comment · Reviewer_KYyp · 2025-08-06
> >
> > Thank you for the additional experiments. I have increased my score. Request you to add these additional details to the revised paper.

---

> ### Author Response · Authors · 2025-08-07
>
> Thank you for your thoughtful feedback and recognition of our work! Your comments have been helpful in guiding us to improve the manuscript. We will incorporate the additional details into the revised version.

---

### Note · Authors · 2025-08-14

We thank all reviewers for their efforts. Their constructive comments and thoughtful feedback have significantly improved our manuscript. We also appreciate their recognition of our work, particularly regarding computational efficiency and effectiveness, which has greatly encouraged us.

Our minimax-remover offers the following advantages:

* **Novelty.** We propose a novel two-stage training strategy. In the first stage, we construct a remover with contrastive tokens that achieves reasonable object removal capability. In the second stage, we use minimax optimization to tame bad noise, enhance robustness, and eliminate usage of CFG.

* **Fast Inference.** Our minimax-remover discards CFG and requires only 6 sampling steps during inference. On a single RTX 4090 GPU, it takes 24 seconds to process a 480P video with 81 frames.

* **Excellent Performance.** Benefiting from minimax optimization, our remover not only effectively removes objects from videos but also seamlessly completes masked regions with coherent background content.

In addition, the reviewer discussion stage further helped us clarify the some points:

* **Motivation and effectiveness of GPT-O3 usage.** Using 100 cases (50 successes and 50 failures) annotated by human experts, we evaluated mainstream LLMs and found that GPT-O3 achieved the best performance, with a 95% overlap with human judgments. This supports the reliability of GPT-O3 in evaluating removal results.

* **Performance on multiple objects and dynamic backgrounds.** We added results for these challenging cases and found our model continues to perform well.

* **Noise patterns and stage-1 success rate.** Using different random seeds in stage-1, we found only 65.56% of cases had consistent success across all seeds, much lower than the individual success rates (71.11%, 72.22%, 75.56%). This shows that success or failure is related to random seeds.

*Moreover, we will include more details that required by reviewers (e.g., training details of stage-1 with contrastive tokens) and provide additional visual comparisons, covering both success and failure cases, in the appendix. We will correct typos and broken links, and add a discussion of broader impacts.*

We believe these revisions substantially strengthen the paper in both empirical scope and clarity. Once again, we thank the reviewers for their valuable input and recognition, which helped us improve the manuscript and motivated us to continue advancing AIGC research.

---

### Decision · Program_Chairs · 2025-09-17

**Decision:**

Accept (poster)

**Comment:**

This paper proposes a two-stage method for video object removal. All the reviewers opine that the paper should be accepted and thus I am recommending acceptance.